# Structural Bias Beyond Homophily: A Study of Fairness in Link Prediction

## Abstract

Graph link prediction (LP) plays a critical role in socially impactful applications such as job recommendation and friendship formation, making fairness a critical concern in this task. While many fairness-aware methods manipulate graph structures to mitigate prediction disparities, the topological biases inherent to social graphs remain poorly understood and are consistently conflated with homophily alone. In this work, we study the relationship between structural biases and fairness outcomes in LP. To this end, we formalize a taxonomy of topological bias measures and introduce a graph generation method producing a diverse corpus of synthetic graphs with controlled structural properties. Using this corpus, we show empirically that fairness outcomes are strongly correlated with graph topology, and that current fairness-aware methods remain sensitive to structural biases beyond homophily. These findings highlight the need for structurally grounded evaluations in fair graph learning.

## 1    Introduction

Graphs are increasingly used in machine learning to model complex interactions between individuals, and the past years have seen the emergence of algorithmic fairness as a central concern across all areas of the field. In this context, researchers and practitioners aim to develop algorithms that do not disproportionately favor or disadvantage specific groups, typically characterized by sensitive attributes such as age, gender, or race. In graph link prediction (LP), fairness takes a particular form. Unlike node classification, where sensitive attributes are defined at the node level, LP operates on pairs of nodes, and the sensitive attribute of an edge derives naturally from those of its endpoints, distinguishing links that connect nodes of the same group from those that connect nodes of different groups. This observation underlies the dyadic fairness paradigm (Li et al., 2021), which seeks to equalize the predicted likelihood of intra- and inter-group links. The connection to homophily (McPherson et al., 2001) is then immediate: homophily precisely measures the tendency of intra-group links to form preferentially, and its well-documented role in producing segregated network structures, e.g., echo chambers, filter bubbles (Pariser, 2011), has made it the natural focus of structural bias research in LP. That said, measuring unfairness through the lens of model outputs has progressively sidelined the study of structural biases inherent in the graphs themselves. Moreover, the limited research connecting graph topology and fairness remains overly focused on homophily. Other forms of structural bias, such as differences in centrality, neighborhood diversity, or information flow across groups, may play an equally critical role, yet remain largely unexamined. This problem is further compounded by the scarcity of publicly available real-world graphs, which obscures the diversity of structural configurations that fairness methods will encounter in practice. Without careful attention to these properties, fairness interventions risk targeting observable disparities rather than their structural origins, limiting their generalizability across network topologies.

In this work, we study the relationship between structural biases and fairness outcomes in LP. We formalize a taxonomy of topological bias measures that extends well beyond homophily, and introduce a parametrizable graph generation process that produces synthetic graphs with controlled structural properties, validated against real-world networks. Crucially, because we control the generative process, we are able to systematically explore a wide range of structural configurations. This allows us to go beyond purely correlational analyses and study how variations in these controlled parameters relate to changes in fairness outcomes

across different bias regimes. Using this approach, we show that fairness results are strongly determined by graph topology, and that current fairness-aware methods remain sensitive to structural biases beyond homophily — including when homophily itself is held constant. The main contributions of this work are:

1. A formalization and categorization of structural biases in graphs, covering topological and flow-based measures beyond homophily (Section 3).

2. A parametrizable graph generation process that synthesizes graphs exhibiting a wide range of structural biases across common link prediction settings, validated against real-world networks (Section 3). We also release the code[1] and the generated graph dataset[2] and introduce a new scientific collaboration graph with gender as a sensitive attribute.

3. An empirical analysis showing that fairness outcomes in LP are strongly determined by graph topology, and that fairness-aware methods remain sensitive to structural biases beyond homophily even when homophily is held constant (Section 4).

## 2 Related Works

### 2.1 Linking Graph Topology and Fair Link Prediction

Fairness in graph learning has been pursued through structural modifications of various kinds: altering the adjacency matrix (Loveland et al., 2022; Spinelli et al., 2021), adjusting random-walk transition probabilities (Rahman et al., 2019; Khajehnejad et al., 2022), or modifying GNN aggregation mechanisms (Dai & Wang, 2021). These approaches all operate, explicitly or implicitly, by reweighting edges according to the sensitive attributes of the nodes they connect. The relationship between topological bias and predictor fairness has nonetheless received limited formal attention. Notable exceptions include works addressing homophily (Laclau et al., 2021; Li et al., 2022), frame structural bias as information access (Jalali et al., 2020; Dong et al., 2022; Arnaiz-Rodriguez et al., 2023), or consider neighborhood heterogeneity (Chen et al., 2022). Broader structural biases have long been discussed outside the learning framework (Borgatti et al., 1998), and recent work quantitatively relates structural features to predictive performance, though without a fairness perspective (Malitesta et al., 2023). Meanwhile, a parallel line of work has sought to refine the very notion of homophily. Zheng et al. (2024b) disentangle graph homophily into label, structural, and feature components, showing that conventional metrics fail to fully account for GNN behavior. Zhang et al. (2025) identify social homophily as a root cause of unfairness in GNNs, while Wang et al. (2022a) study graph embedding under biased observations. These contributions confirm that homophily is a richer and more complex phenomenon than typically assumed, yet homophily remains the organizing concept through which structural bias is examined.

### 2.2 Graph Generation

Classical graph generation models have long provided foundational tools for replicating key network properties such as community structure, degree distribution, and scale-free topology (Holland et al., 1983; Newman, 2003; Barabási & Albert, 1999). These models were not designed with fairness considerations in mind and generally lack mechanisms to control parameters such as homophily or sensitive attribute imbalance. Fairness-aware approaches have therefore emerged, often relying on graph cloning techniques (Zhu et al., 2022; Zheng et al., 2024a) that replicate an existing graph. While effective for privacy-related purposes, these methods depend on a source graph and offer limited flexibility in generating networks with arbitrary or customizable structural properties. The Biased Preferential Attachment (Stoica et al., 2018) supports homophily tuning during graph growth, and the Contextual Stochastic Block Model (Deshpande et al., 2018) allows community-correlated node features, but both rely on a single attachment mechanism, limiting the structural diversity of the generated graphs. Learning-based generators such as GenCAT (Maekawa et al., 2023) synthesize graphs with controllable community structures and attribute distributions, but their lack

---

[1]https://anonymous.4open.science/r/TMLR_submission-C55C
[2]https://zenodo.org/records/20143150

of interpretable parameter control makes them less suited to the kind of controlled structural intervention our analysis requires.

## 2.3 Evaluation in Fair Graph Learning

Evaluation efforts in graph learning have primarily relied on a small number of real-world datasets (Shchur et al., 2018; Li et al., 2023; Delarue et al., 2024), which limits the ability to assess model behavior across diverse structural configurations. The use of synthetic data for evaluation was introduced for node classification by Maekawa et al. (2022), who showed that controlled graph generation enables more reliable model comparisons. Their framework varies class imbalance, homophily, attribute distribution, and graph order, but does not capture the finer-grained structural biases that fairness-oriented methods specifically aim to address.

### Position of This Work

The works reviewed above share a common limitation: structural bias in graphs is treated primarily through the lens of homophily, whether in the design of fairness interventions, the construction of generative models, or the choice of evaluation datasets. Even recent efforts to refine the notion of homophily remain within this conceptual perimeter. In this work, we take a different perspective and ask whether the fairness of LP methods depends on structural properties that homophily alone cannot capture. To this end, we combine a broader taxonomy of structural bias measures with a generative process that allows a wide range of bias configurations. This approach enables us to move beyond correlational observations and assess the specific contribution of each structural bias to fairness outcomes.

## 3 Structural Biases and Fairness in Link Prediction

We consider undirected graphs $\mathcal{G} = (\mathcal{V}, \mathcal{E})$ where $\mathcal{V}$ is the set of nodes and $\mathcal{E} \subset \mathcal{V} \times \mathcal{V}$ is the set of edges. Each node is assigned a sensitive attribute via a function $S : \mathcal{V} \to \{0, 1\}$, where $S(v) = 0$ denotes a non-sensitive node and $S(v) = 1$ denotes a sensitive node. We study how structural properties of $\mathcal{G}$ determine fairness outcomes in LP by combining a taxonomy of structural bias measures with a controlled graph generation process, enabling the exploration of a wide range and diversity of biases across specific use cases.

We first formalize the structural biases we consider, then describe the generative process and its validation against real-world networks, and finally present the experimental setup and the hypotheses we test.

### 3.1 Structural Bias Measures

To systematically characterize structural disparities between sensitive groups, we propose a unified taxonomy of bias measures (Table 1) that integrates classical graph metrics (Borgatti et al., 1998) with recent fairness-oriented ones. The taxonomy distinguishes node-level measures, which capture local structural disparities, from group-level measures, which reflect global patterns of separation and influence. Within each level, measures are further classified according to whether they capture topological or flow-based properties of the graph. Formal definitions of each measure are provided in Appendix A.

### 3.1.1 Node-level Measures

Node-level measures reflect different dimensions of structural prominence and neighborhood composition. Centrality measures such as closeness, betweenness, and eigenvector centrality (prestige) quantify a node's influence within the network. Neighborhood-based metrics such as $k$-hop degrees, ego-network density and heterogeneity describe local connectivity and diversity. Each captures a distinct aspect of structural disadvantage: differences in degree translate into unequal chances of forming new links across groups (Subramonian et al., 2023), while heterogeneity reflects whether nodes connect primarily within or across group boundaries (Masrour et al., 2020).

To quantify bias from these measures in a unified way, we define the normalized difference in expected values across sensitive groups.

**Definition 1** (Node-level Disparity). Let $\mathcal{G} = (\mathcal{V}, \mathcal{E})$ be a graph, $S \in \{0, 1\}$ a sensitive attribute, and $M : \mathcal{V} \to \mathbb{R}$ a node-level measure. The bias associated with $M$ is defined as:

$$\omega_M(\mathcal{G}) = \frac{\mathbb{E}[M(\mathcal{V}) \,|\, S = 0] - \mathbb{E}[M(\mathcal{V}) \,|\, S = 1]}{\mathbb{E}[M(\mathcal{V})]}.$$

Positive values indicate a structural advantage for the non-sensitive group, while $\omega_M(\mathcal{G}) = 0$ corresponds to parity.

Beyond static topology, we include flow-based measures such as effective resistance measures (ISOLATION, DIAMETER and CONTROL) (Arnaiz-Rodriguez et al., 2023), which model nodes' information dissemination via random walks. Two groups may exhibit similar average degree yet differ substantially in effective resistance, reflecting asymmetries in accessibility or resilience in information flow that static measures cannot capture.

### 3.1.2 Group-level Measures

Group-level measures capture global structural disparities between sensitive groups. Assortativity quantifies homophily, that is, the tendency of nodes to connect within their own group. The average mixed distance reflects how closely nodes from different groups are connected, which constrains cross-group link formation. The power law exponent ratio compares degree distributions across groups, indicating whether one group disproportionately concentrates hub positions (Barabási & Albert, 1999). Information unfairness (Jalali et al., 2020) measures asymmetries in how information propagates across groups, capturing dynamic biases that go beyond static structure.

Together, these measures form a coherent framework that covers the principal dimensions along which graph topology can be structurally biased. Their diversity is precisely what motivates the controlled generation process described next: studying their individual contributions to fairness outcomes requires a setting where each can be varied independently.

Table 1: Unified taxonomy of structural bias relevant to fairness. Measures are classified by their scope (local vs. global) and whether they capture topological or flow-based aspects of the graph. Notably, measures introduced in recent fairness-oriented work are exclusively flow-based.

| Node-based measures | A node considered is favored if... | Scope | Type |
|---|---|---|---|
| CLOSENESS | It is close to all nodes | Shortest paths | Topology |
| BETWEENNESS | Many shortest paths go through it | Shortest paths | Topology |
| PRESTIGE | Its neighbors have high eigencentrality | Graph | Topology |
| DEGREE | It has many neighbors | Neighbors | Topology |
| CONSTRAINT | Its neighbors have many neighbors | 2-hop Neighbors | Topology |
| DENSITY | Its neighbors are not clustered | Neighbors | Topology |
| HETEROGENEITY | Its neighbors are diverse w.r.t sensitive attr. | Neighbors | Topology |
| EFFECTIVE RESISTANCE | It has strong information flow | Random walks | Flow |
| **Group measures** | **Groups are evenly favored if...** | **Scope** | **Type** |
| ASSORTATIVITY | They are interconnected a lot | All graph edges | Topology |
| AVG MIXED DIST | They are close to each other (average) | Shortest paths | Topology |
| POWER EXP | They have same degree distribution | All graph edges | Topology |
| INFO UNFAIRNESS | They have same flow distribution | Random walks | Flow |

### 3.2 Controlled Graph Generation for Structural Intervention

To study the effect of structural biases on fairness outcomes, we extend the Barabási–Albert (BA) model Barabási & Albert (1999) to produce graphs whose structural properties can be controlled and varied

systematically. The standard BA model incrementally adds nodes that attach preferentially to high-degree nodes, reproducing scale-free degree distributions, but provides no mechanism to incorporate sensitive attributes, control homophily, or modulate community structure and degree heterogeneity.

We address these limitations through four modular extensions, described below and summarized in Algorithm 1.

### Sensitive Attribute Assignment

A parameter $\alpha \in (0, 1)$ controls the fraction of non-sensitive nodes, assigning sensitive attributes independently of node arrival order. This ensures that group imbalance is a free parameter, decoupled from the growth dynamics of the graph.

### Homophily Control

Attachment probabilities are adjusted to favor connections between nodes sharing the same sensitive attribute, governed by a parameter $\beta \geq 0$. The probability that a new node $v_{\text{new}}$ attaches to an existing node $v$ is given by:

$$P_{\beta,S}(v \mid S(v_{\text{new}})) \sim \deg_{\mathcal{G}}(v) + \mathbf{1}\{S(v) = S(v_{\text{new}})\} \cdot (e^{\beta} - 1), \tag{1}$$

where $\beta = 0$ recovers standard preferential attachment and increasing $\beta$ progressively favors intra-group connections.

**Proposition 2.** For any fixed $\alpha$ and graph order $n$, the expected assortativity $\mathbb{E}[\omega_{\text{ASSORT}}(\mathcal{G})]$ is monotonically increasing in $\beta$.

*Sketch.* At each growth step $t$, the probability that a newly created edge is intra-group is strictly increasing in $\beta$, as shown in Appendix B.1. Since assortativity is an increasing function of the proportion of intra-group edges, the result follows by linearity of expectation over growth steps. $\square$

This property is what makes the generative process suitable for structural intervention: because $\beta$ controls assortativity monotonically and $\alpha$ is set independently, fixing $\alpha$ and varying $\beta$ constitutes a direct intervention on assortativity, allowing for a robust analysis of its effect.

### Community Structure via Anchor Nodes

New nodes enter through a designated anchor node and preferentially connect within its local neighborhood, modeling community-driven attachment. A new node $v_{\text{new}}$ first connects to an anchor $u$, then forms additional edges with nodes in $u$'s ego network, with attachment probability $P_{\text{ego}}(\cdot \mid u)$ decreasing with hop distance. This reflects the social phenomenon whereby individuals join communities through a common acquaintance.

### Adjusting Number of Connections

Rather than attaching a fixed number of edges $m$ per new node, we sample the number of connections from a Gamma distribution:

$$m' \sim \Gamma\left(\gamma, \frac{m}{\gamma}\right),$$

where $m$ remains the expected number of edges and $\gamma$ controls the variance. Lower values of $\gamma$ produce greater variability in node connectivity, reproducing the skewed patterns observed in empirical graphs. The Gamma distribution is a natural choice here, as it is defined on $\mathbb{R}_{>0}$ and provides a flexible two-parameter family. It interpolates between near-deterministic and highly dispersed regimes through $\gamma$ alone while keeping the mean fixed.

---

**Algorithm 1:** Extended BA Graph Generation

---

**Input:** Graph order $n$, group imbalance $\alpha$, homophily parameter $\beta$, expected degree $m$, *anchor* flag, node degree probability $P_{m'}$, ego network attachment $P_{\text{ego}}$

**Output:** Generated graph $\mathcal{G} = (\mathcal{V}, \mathcal{E})$ and sensitive attributes $S$

---

**1** Initialize $\mathcal{V} \leftarrow \{1, \dots, n\}$;
**2** $S \leftarrow \text{AssignSensitive}(\mathcal{V}, \alpha)$;
**3** Initialize $\mathcal{G} = (\mathcal{V}', \mathcal{E}) \leftarrow$ star graph with $\mathcal{V}' = \{1, \dots, m\}$;

**4** **for** $v_{new} \leftarrow m + 1$ **to** $n$ **do**
**5**      $\mathcal{V}' \leftarrow \mathcal{V}' \cup \{v_{\text{new}}\}$;
**6**      Sample $m' \sim P_{m'}$ and set $m' \leftarrow \min(m', v_{\text{new}} - 1)$ // Defining $v_{\text{new}}$ degree $m'$;
**7**      **if** *anchor* **then**
**8**          Sample anchor $u \sim P_{\beta,S}(\cdot \mid S(v_{\text{new}}))$ // Defining $v_{\text{new}}$ anchor node from its sens. attr.;
**9**          $N(v_{\text{new}}) \leftarrow \text{SampleDistinct}(m', P_{\text{ego}}(\cdot \mid u))$// Sampling $v_{\text{new}}$ neighbors from anchor ego-network;
**10**          $N(v_{\text{new}}) \leftarrow N(v_{\text{new}}) \cup \{u\}$;
**11**      **else**
**12**          $N(v_{\text{new}}) \leftarrow \text{SampleDistinct}(m', P_{\beta,S}(\cdot \mid S(v_{\text{new}})))$// Sampling $v_{\text{new}}$ neighbors from $\mathcal{V}$;

**13**      $\mathcal{E} \leftarrow \mathcal{E} \cup \{(v_{\text{new}}, v) \mid v \in N(v_{\text{new}})\}$;
**14**      $\mathcal{G} \leftarrow (\mathcal{V}', \mathcal{E})$;

**15** **return** $(\mathcal{G}, S)$;

---

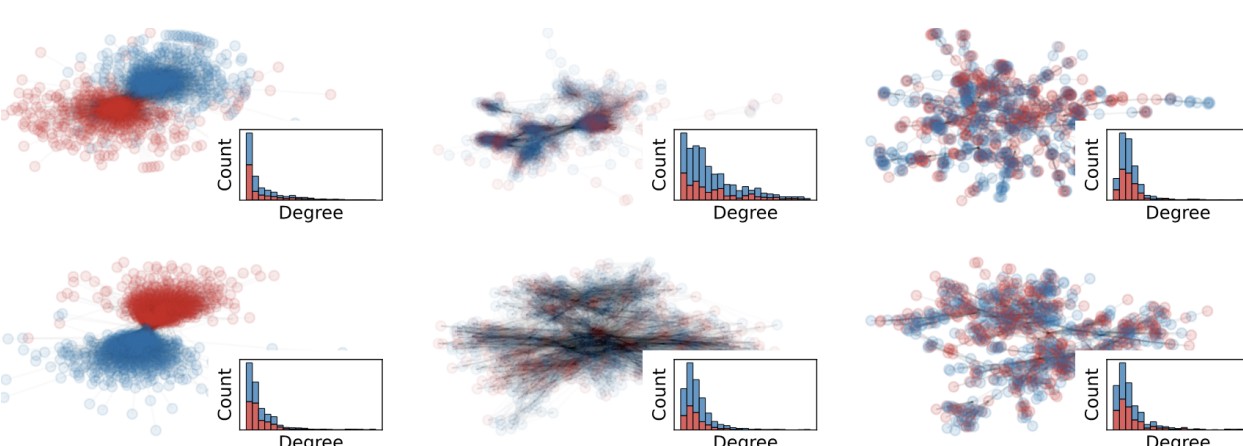

Figure 1: Comparison between real (top) and generated (bottom) graphs with degree distributions, for *Opinion* (left), *Friendship* (center), and *Collab* (right) use cases. Red and blue colors indicate respectively sensitive and non-sensitive nodes. All generation parameters are fitted on real datasets.

### 3.3 Validation Against Real-World Networks

To ensure that the generated graphs reflect structural patterns encountered in practice, we calibrate the generative process against three real-world networks with distinct fairness contexts: Polblogs Adamic & Glance (2005) (*Opinion*), Facebook ego-networks Leskovec & Mcauley (2012) (*Friendship*), and a co-authorship network (*Collab*). These cases involve different sensitive attributes: political affiliation for *Opinion*, gender for *Friendship* and *Collab*. Further details on each dataset are provided in Appendix B.2.

The parameters $n$, $m$, $\gamma$, and the weights assigned to $k$-hop neighbors are calibrated to match the structural properties of each real-world dataset, via a grid search over each parameter's range of plausible values. Table 2 summarizes the resulting configurations. Figure 1 compares degree distributions and attribute-aware

Table 2: Generation configurations for each use case. Anchor weights indicate attachment probabilities for 1-hop, 2-hop, and 3+-hop neighbors respectively. $\deg(u)$ refers to the degree of the anchor node.

|  | *Opinion* | *Friendship* | *Collab* |
|---|---|---|---|
| Source of homophily | Sampled neighbors | Anchor node | Anchor node |
| Anchor weights | – | $10^3/2/1$ | $1/0$ |
| $m'$ | $\Gamma(m{=}14,\ \gamma{=}0.08)$ | $0.55 \cdot \deg(u) + 3$ | $\Gamma(m{=}3,\ \gamma{=}1)$ |

structural patterns of generated and real graphs, showing close alignment. Further quantitative validation is provided in Appendix B.3.

Once the global generation parameters are fixed, $\alpha$ and $\beta$ can be varied freely to produce graphs spanning a wide range of structural bias configurations.

### 3.4 Experimental Setup and Hypotheses

We introduced a comprehensive taxonomy of structural graph biases, along with a synthetic graph generation framework that enables diverse variation of bias configurations. This setting enables the study of LP algorithms under a substantially broader spectrum of graph biases than commonly considered in prior work.

For each of the three real-world use cases, synthetic graphs are generated by varying the group imbalance parameter $\alpha$ and the homophily parameter $\beta$ over a $32 \times 32$ grid, resulting in a corpus of 1,024 graphs per use case and per random seed.

In the following, LP is formulated as a recommendation task in which each node is recommended $k = 10$ potential new connections. Edges are randomly split into training (80%) and test (20%) sets. Models are trained on the training graph and evaluated on the test graph by ranking candidate edges according to their predicted existence probabilities. Reported results are averaged over five independent train/test splits.

The evaluated models are divided into two categories: classical LP methods and fairness-aware methods. The classical models consist of embedding-based approaches spanning the three main families of graph representation learning for link prediction: random-walk-based methods with Node2Vec (Grover & Leskovec, 2016) (N2V), matrix factorization methods with Singular Value Decomposition (SVD) (Halko et al., 2011), and neural-network-based methods with Graph Convolutional Networks (GCN) (Kipf & Welling, 2016). The fairness-aware methods include DeBayes (Buyl & De Bie, 2020), CrossWalk (Khajehnejad et al., 2022), FairAdj (Li et al., 2021), FLIP (Masrour et al., 2020), UGE (Wang et al., 2022b), and MORAL (Mattos et al., 2026).

Each graph is represented by the set of structural bias measures introduced in our taxonomy. Model performance is evaluated using HitRank and AUC, while fairness is assessed through Statistical Parity and Equal Opportunity (Masrour et al., 2020).

This experimental protocol constitutes the empirical basis for testing the following hypotheses:

**H1**: Fairness outcomes of classical LP methods are strongly determined by the structural bias measures of the underlying graph.

**H2**: Fairness LP models are only as effective as their ability to address the right source of topology bias.

**H3**: Controlling for assortativity, structural biases beyond homophily have no significant effect on the fairness outcomes of fairness-aware methods.

Hypothesis H3 is formulated as a null hypothesis. Our analysis aims to reject it by showing that structural biases beyond homophily exert an independent effect on fairness outcomes even when assortativity is held constant.

# 4 Results

The three hypotheses formulated in Section 3.4 are tested in turn. We first validate that the generated corpus covers a diverse range of structural configurations, then assess the relationship between structural bias and fairness outcomes for classical methods (H1), examine whether fairness-aware methods exhibit comparable sensitivity (H2), and finally test whether this sensitivity persists when assortativity is held constant (H3). All models achieve strong predictive performance across the corpus (see Appendix D), ensuring that observed fairness differences are not attributable to weak predictive quality.

## 4.1 Corpus Validation: Coverage of the Structural Bias Space

Before testing the hypotheses, we verify that the generated corpus spans a sufficiently diverse range of structural configurations. Figure 2 shows a principal component analysis of the bias measures from our taxonomy, computed over the full corpus for each use case. The corpus spans a wide range of biases, as evidenced by the dispersion of points and the fact that no principal axis (detailed in Appendix B.3) is dominated by a single bias measure.

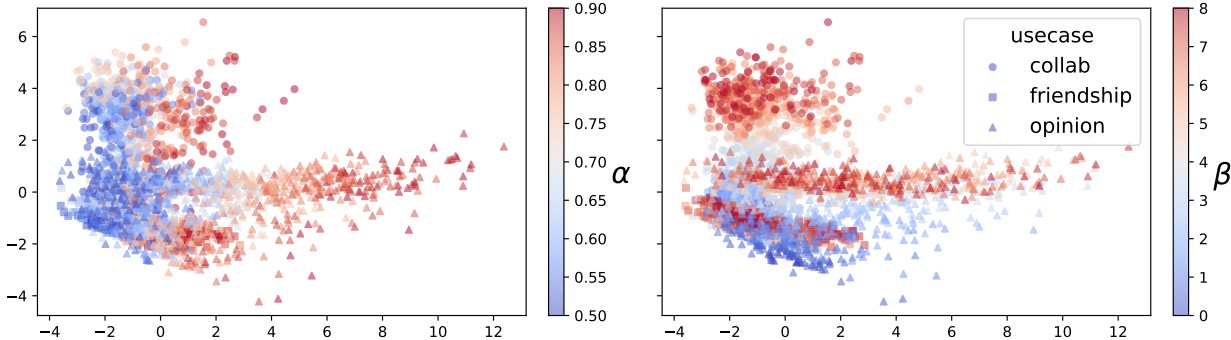

Figure 2: PCA of structural bias measures over the generated corpus, for each use case. Left: colored by group imbalance $\alpha$. Right: colored by homophily parameter $\beta$. Each point represents a generated graph. Axis definitions are provided in Appendix B.3.

## 4.2 Testing H1: Structural Bias as a Determinant of Fairness Outcomes

In this section, we focus on classical link prediction models. To assess the extent to which structural biases determine fairness outcomes, we train Random Forest regressors to predict fairness scores from the bias measures defined in our taxonomy. We report the resulting $R^2$ scores, as well as the proportion of $R^2$ reached by ASSORTATIVITY alone.

Figure 3 (left) shows that fairness outcomes are strongly determined by graph topology across all use cases and models, with $R^2$ values consistently above 0.90. This indicates that the proposed bias taxonomy captures structural information that is genuinely predictive of model fairness, and that the fairness scores of standard models are largely predictable from graph-level structural features. Furthermore, ASSORTATIVITY alone accounts for a substantial share of the $R^2$ across all three use cases, lending empirical support to the central role this measure plays in driving unfairness in link prediction tasks.

We also observe that $R^2$ values are systematically lower for regressions targeting Equal Opportunity compared to Statistical Parity. This is consistent with the fact that EO is conditioned on the existence of edges, exposing more complex structural dependencies that are harder to capture through the proposed features alone.

In Figure 3 (right), we present examples of structural biases importance scores in computed regressions. In the *Opinion* setting, ASSORTATIVITY dominates regardless of the embedding model, indicating that the relationship between structural bias and fairness is stable across methods when the topology is sufficiently simple. In *Friendship*, the importance of heterogeneity grows substantially, reflecting the more complex

| Use case | Model | $R^2$ SP | $R^2$ EO | Assort. $\%R^2$ SP | Assort. $\%R^2$ EO |
|----------|-------|----|----|----|----|
| *Opinion* | GCN | 1.0 | 0.91 | 97 | 92 |
| | N2V | 1.0 | 0.93 | 97 | 89 |
| | SVD | 0.99 | 0.91 | 93 | 95 |
| *Friend.* | GCN | 0.99 | 0.97 | 89 | 96 |
| | N2V | 0.99 | 0.97 | 88 | 95 |
| | SVD | 0.99 | 0.97 | 81 | 97 |
| *Collab* | GCN | 1.0 | 0.94 | 95 | 97 |
| | N2V | 1.0 | 0.96 | 95 | 97 |
| | SVD | 0.99 | 0.93 | 83 | 97 |

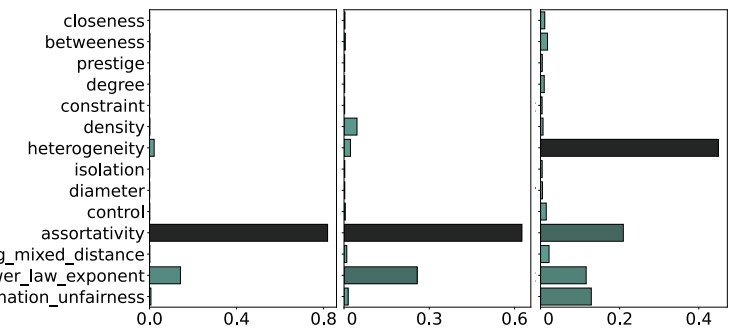

Figure 3: Left: $R^2$ scores and proportion of $R^2$ reached by ASSORTATIVITY alone, for regression of structural bias measures on fairness metrics across all use cases. Right: Feature importance scores from the structural bias regression, for *Opinion* SP with N2V (left), *Opinion* SP with SVD (middle), and *Friendship* SP with N2V (right).

interplay between local neighborhood structure and fairness outcomes in denser graphs. The full set of feature importance profiles across use cases and models (reported in Appendix E) further supports the conclusions drawn from the $R^2$ analyses. While ASSORTATIVITY frequently emerges as the most influential bias measure, several settings exhibit different patterns. In some cases, another structural bias becomes predominant, such as HETEROGENEITY for $SP$ regression with $GCN$ model in *Friendship* use case. In others, importance is distributed across multiple bias measures, as observed for $EO$ regression with $N2V$ model in *Opinion* use case.

These results support H1: fairness outcomes of classical LP methods are strongly determined by the structural bias measures of the underlying graph, with assortativity playing a central but not exclusive role.

### 4.3 Testing H2: Fairness-Aware LP Methods Under Structural Bias

For H2 and H3, the graphs are generated on a $16 \times 16$ grid because training all the fair LP models on the entire corpus would be too computationally expensive.

The high structural dependence observed in H1 raises the question of whether fair models mitigate this dependence. Figure 4 displays the distribution of fairness and performance outcomes across the corpus generated for each use case (256 graphs) and each model. Across all models and use cases, fairness outcomes exhibit significantly higher variance than performance outcomes. This asymmetry implies that slight variations in graph topology, while having a more limited effect on accuracy, can lead to very different fairness outcomes.

In *Opinion*, the quasi-bipartite node structure makes intergroup edges both structurally rare and essential for predicting cross-group links. Walk-based models get trapped within hubs and fail to classify inter-hub test edges. DeBayes, MORAL, and FLIP avoid this trap through distinct mechanisms: DeBayes reduces the weighting of hub nodes via its a priori degree correction, thereby allowing inter-hub candidates to emerge; MORAL dedicates a separate encoder to intergroup edges, explicitly learning the rare inter-hub signal; FLIP's adversarial loss imposes community-invariant embeddings, forcing the generator to transcend hub boundaries. All three models achieve higher accuracy and lower SP (see Figure 31 in Appendix F).

In *Friendship*, the absence of a dominant source of structural bias (balanced degree distributions, intertwined communities, ASSORTATIVITY capped at 0.5) makes the trade-off between performance and fairness less pronounced, but not nonexistent. FLIP and CrossWalk achieve the best fairness outcomes, at the expense of AUC and Hit@10, respectively.

In *Collab*, the primary source of structural bias lies in the maximum average distance between groups, which structurally limits the flow of information between them, leading walk-based models to produce representations that fail to accurately capture inter-group relationships. DeBayes is the only model whose mechanism

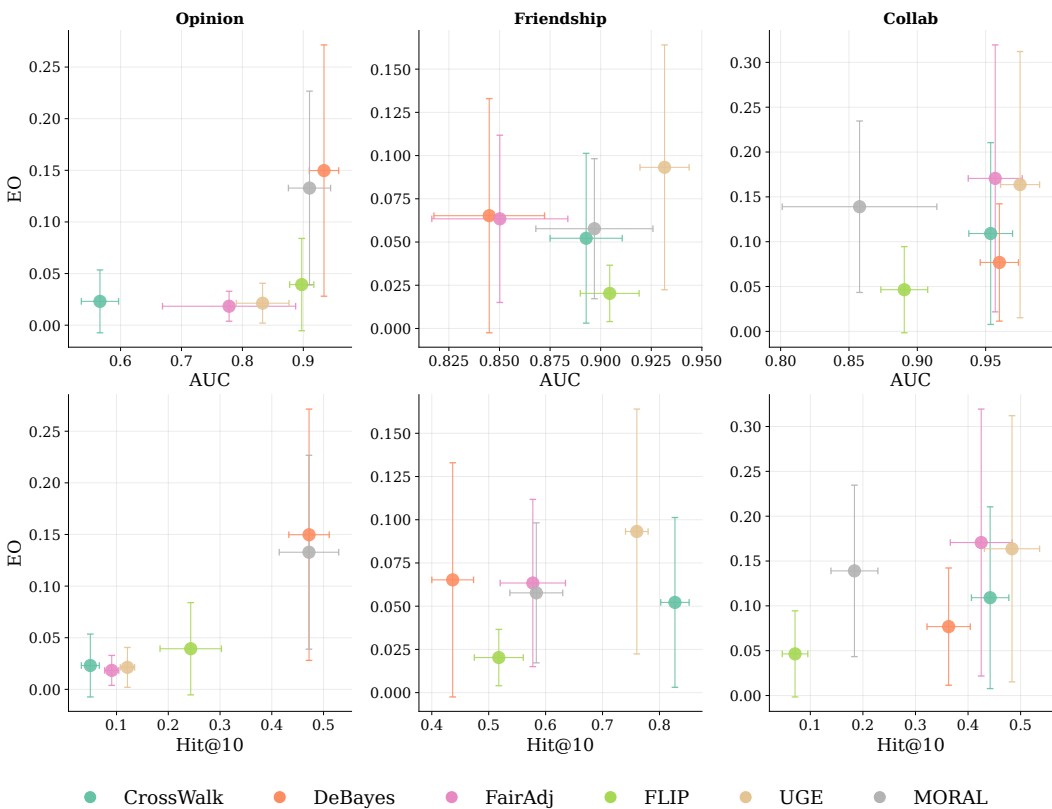

Figure 4: Distribution of fairness and performance outcomes across the synthetic graph corpus, for each fair LP model and use case. Each point represents the mean over all synthetic graphs; error bars show ± standard deviation.

consists of inverting the prior distribution of degrees at the time of evaluation and neutralizing this confounding factor without altering the graph structure, resulting in a true improvement over EO and SP. FairAdj operates on a reweighted adjacency matrix that modifies the original graph structure, while MORAL fails to leverage the full graph by restricting each model to a subset of edges, both resulting in an unnecessary performance cost.

These results support H2: the key factor determining the effectiveness of a fairness-aware LP model is not the strength of the intervention designed to ensure equity in itself, but its ability to address the actual source of bias in the graph.

### 4.4 Testing H3: Beyond-Homophily Biases at Fixed Assortativity

The results of the previous sections show that assortativity plays a central role in determining fairness outcomes, but leave open the question of whether other structural biases have an independent effect once homophily is accounted for. H3 posits that they do not.

To test this, we compute the partial Spearman rank correlation to isolate the influence of structural biases from the effects of ASSORTATIVITY. First, the structural biases, the model outcomes, and the ASSORTATIVITY are transformed into ranks to handle potential non-linearities while maintaining monotonicity. We then perform an OLS regression on these ranks to extract the residuals, the variance in the bias and the model outcomes that remain unexplained by ASSORTATIVITY. The partial correlation coefficient, $\rho$, measures the specific association between structural biases and the model's output after controlling for the confounding influence of homophily. To account for the grouped structure of our dataset, where multiple models are evaluated on the same graph instance, we employ a block-permutation test. By permuting graph labels

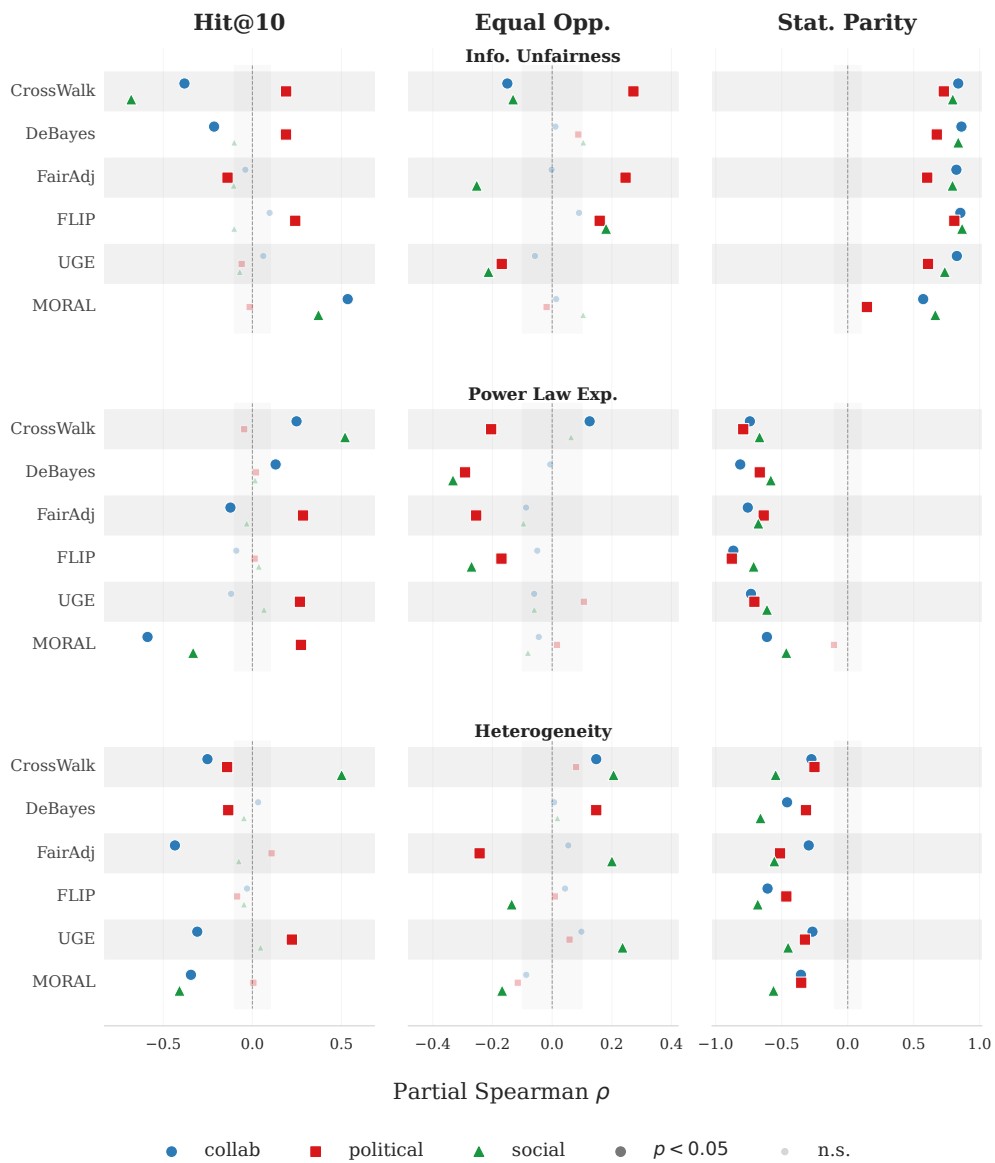

Figure 5: Partial Spearman $\rho$ between the top-3 structural biases and fairness/performance outcomes across models and use cases. Marker opacity encodes statistical significance ($p < 0.05$).

rather than individual observations, we treat each graph as a single unit, thereby preserving the intra-graph dependencies and the underlying data topology. This procedure allows us to construct a robust empirical null distribution for our partial Spearman coefficients, ensuring valid p-value inference despite the non-i.i.d. nature of the samples.

The three structural biases selected beyond ASSORTATIVITY capture complementary dimensions of structural inequality: the concentration of high-degree nodes within one group (POWER EXP), the diversity of individual node neighborhoods across group boundaries (HETEROGENEITY), and the asymmetry in cross-group information flow (INFO UNFAIRNESS). The other biases are in Appendix **??**.

As shown in Figure 5, the three structural biases have varying effects across metrics, models, and use cases once the assortativity has been controlled. The SP consistently yields the highest and most significant Spearman partial correlations for all three types of bias and all three datasets. This indicates that graphs exhibiting greater structural inequality consistently show poorer statistical parity, regardless of the method

used to predict the links. In contrast, EO exhibits correlations of varying signs : the direction and magnitude of $\rho$ vary considerably depending on both the models and the usecase, with no structural bias producing a stable and uniform effect. This inconsistency highlights the fact that EO results from the interaction between the graph structure and the inductive biases of each model, such that a model that performs well in one use case may fail in another with different structural properties, underscoring the need to evaluate equality of opportunity by taking into account both the model and the topology.

These results reject H3: even when ASSORTATIVITY is constant, other structural biases exert a statistically significant effect on fairness outcomes, demonstrating that homophily alone is insufficient to characterize the structural conditions under which fairness methods operate.

## 5 Conclusion

This work set out to examine whether the fairness of link prediction methods depends on structural properties of the underlying graph that go beyond homophily. Fairness outcomes are unambiguously and strongly determined by graph topology across all use cases and models considered, and this dependence persists even when homophily is held constant: bias measures such as heterogeneity, information unfairness, and power law exponent ratio exert an independent and statistically significant effect on the fairness of both classical and fairness-aware methods. None of the evaluated approaches achieves robustness to this broader set of structural biases.

These findings rest on two methodological contributions. The first is a taxonomy of structural bias measures that covers topological and flow-based properties at both node and group levels, providing a more complete characterization of the structural conditions under which fairness methods operate. The second is an iterative and modular graph generation framework that reproduces several classical link prediction use cases while enabling controlled assortativity and the exploration of a broad range of structural bias configurations. The empirical analysis draws its strength from this second contribution: rather than relying solely on correlational observations over a fixed collection of graphs, our approach systematically explores controlled variations of assortativity and structural configurations across a corpus of more than one thousand graphs per use case, validated against real-world networks.

This study has several limitations. The generative process is currently limited to binary sensitive attributes and to undirected graphs, which restricts its applicability to settings involving more complex demographic structures or directed social networks. Finally, while the controlled generation process goes beyond purely correlational analysis, a fully formal causal treatment of the relationships studied here would require additional assumptions and methodology that fall outside the scope of this work.

Future work could extend the taxonomy and the generative process to multi-valued and multivariate sensitive attributes, enabling the study of intersectional bias in graph structures. A natural next step would also be to use the generated corpus as a tool for the design of fairness-aware methods, rather than solely for their evaluation: a method trained or regularized on structurally diverse graphs might exhibit the kind of robustness that current approaches lack. More broadly, the perspective developed here, namely that structural bias is a multidimensional phenomenon that cannot be reduced to homophily alone, applies equally to other graph learning tasks such as node classification or graph-level prediction, and we hope this work encourages its adoption beyond the specific setting of link prediction.

### Broader Impact Statement

This work is primarily a diagnostic contribution: it identifies structural conditions under which fairness interventions in link prediction succeed or fail, and provides tools to study them systematically. We do not deploy predictive systems nor introduce new methods that could directly cause harm.

Our findings highlight that current fairness-aware methods remain sensitive to structural biases beyond homophily. This result should be interpreted as a call for more careful evaluation practices in consequential applications such as job recommendation or friendship suggestion, where overlooking structural bias may lead to fairness guarantees that do not generalize across diverse network topologies.

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

## Computing Infrastructure

All experiments were run on a MacBook with an Apple M3 Pro chip, 11-core CPU, and 18GB of unified memory, with the exception of GCN-based experiments and fairness-aware baselines requiring GPU acceleration, which were executed on an NVIDIA Tesla V100 GPU. The software environment was based on Python 3.11.6. Versions of all relevant packages are specified in the provided code.

## A    Formal Definitions of Structural Bias Measures

We detail here the mathematical formalization of structural bias measures defined in Section 3.1.

In the following, we denote $\mathcal{V}_0$ and $\mathcal{V}_1$ the sets of non-sensitive and sensitive nodes, respectively.

More, for $v, v' \in \mathcal{V}$, and for $V \subset \mathcal{V}$:

- $\sigma(v, v')$ is the set of shortest paths between $v$ and $v'$ and $\sigma_{v,v'}$ is their length.

- $\lambda$ is the largest eigenvalue of the adjacency matrix of $\mathcal{G}$ and $x_{max}$ the corresponding eigenvector.

- $\mathcal{G}_V$ is the sub-graph of $\mathcal{G}$ induced by $V$ nodes.

- $N(v)$ is the set of $v$ neighbors.

- $d(\mathcal{G})$ is the classical graph density of $\mathcal{G}$.

### A.1    Node-based Measures

In this section, we provide the formulas for node-based measures. Hence, all measures of bias take a node (denoted by $u$) as input.

$$\text{CLOSENESS}(u) = \frac{|\mathcal{V}| - 1}{\sum\limits_{v \in \mathcal{V}, v \neq u} \sigma_{u,v}}$$

$$\text{BETWEENNESS}(u) = \sum_{v,v' \in \mathcal{V}} \frac{|\{p \in \sigma(v, v') | u \in p\}|}{|\sigma(v, v')|}$$

$$\text{PRESTIGE}(u) = \frac{1}{\lambda} \sum_{v \in N(u)} x_{max,v}$$

$$\text{DEGREE}(u) = |N(u)|$$

$$\text{CONSTRAINT}(u) = \sum_{v \in N(u)} |N(v)|$$

$$\text{DENSITY}(u) = 1 - d(\mathcal{G}_{N(u) \cup \{u\}})$$

$$\text{HETEROGENEITY}(u) = 1 - 2 \left| \left( \frac{1}{|N(u)|} \sum_{v \in N(u)} S(v) \right) - \frac{1}{2} \right|$$

*Effective Resistance Measures Arnaiz-Rodriguez et al. (2023)* The effective resistance is defined $\forall u, v \in \mathcal{V}$ as $\mathcal{R}_{uv} = (e_u - e_v) L^\dagger (e_u - e_v)^T$, where $e_u$ is the unit vector with a 1 value at $u$-th index and zero elsewhere, and $L^\dagger$ denotes the pseudo-inverse of the graph's Laplacian. Then, the *strength* of a node $u$ can be computed with three different ways:

$$R_{tot}(u) = \sum_{v \in V} \mathcal{R}_{uv},$$

$$R_{diam}(u) = \max_{v \in V} \mathcal{R}_{uv},$$

$$B_R(u) = \sum_{v \in N(u)} \mathcal{R}_{uv}.$$

For aggregating these node-based measures, the authors use the following function:

$$\Delta_R(\mathcal{G}) = \left| \frac{1}{|\mathcal{V}_0|} \sum_{u \in \mathcal{V}_0} R(u) - \frac{1}{|\mathcal{V}_1|} \sum_{u \in \mathcal{V}_1} R(u) \right|$$

with $R \in \{R_{tot}, R_{diam}, B_R\}$, defining ISOLATION, DIAMETER and CONTROL, respectively.

## A.2 Graph/Group measures

Here, we give the formula for group measures, which apply directly on $\mathcal{G}$.

$$\text{ASSORTATIVITY}(\mathcal{G}) = \frac{Tr(\mathbf{e}) - \sum_{i,j} \mathbf{e}_{i,j}}{1 - \sum_{i,j} \mathbf{e}_{i,j}},$$

where $\mathbf{e}$ is a matrix of size 2, with $\mathbf{e}_{0,0}$ (resp. $\mathbf{e}_{1,1}$) representing the proportion of edges connecting two sensitive (respectively non-sensitive) nodes, and $\mathbf{e}_{1,0} = \mathbf{e}_{0,1}$ representing each half of the proportion of edges connecting a sensitive and a non-sensitive node.

$$\text{AVG MIXED DIST}(\mathcal{G}) = \frac{1}{|\mathcal{V}_0||\mathcal{V}_1|} \sum_{v,v' \in \mathcal{V}_0 \times \mathcal{V}_1} \sigma_{v,v'}$$

$$\text{POWER EXP} = \frac{\kappa_1}{\kappa_0}.$$

where $\kappa_0$ and $\kappa_1$ denote the exponents of the power-law degree distributions of non-sensitive and sensitive nodes, respectively.

Then Information Unfairness Jalali et al. (2020) can be written as

$$\begin{aligned} \text{INFO UNFAIRNESS} = \max(&\text{dist}(D_{00}, D_{01}), \\ &\text{dist}(D_{00}, D_{11}), \\ &\text{dist}(D_{11}, D_{01})) \end{aligned}$$

where $D_{s_1, s_2} = \{A_{uv} | S(u) = s_1, S(v) = s_2, u \neq v\}$, with $A_{uv}$ being the probability of random walk information flowing from node $u$ to node $v$.

# B  Generation

## B.1  Proof of Proposition 1

*Proof.* The proof consists of three steps: first, we show that the probability of forming an intra-group edge at each growth step is increasing in $\beta$; second, we generalise this result to the expected global proportion of intra-group edges; finally, we show that the assortativity is increasing in this proportion.

**Step 1: monoticity of $p^t_{\mathbf{intra}}(\beta)$.**  At step $t$ of the growth process, a new node $v_{\text{new}}$ attaches to $m'$ existing nodes. Let $\mathcal{V}^t_s = \{v \in \mathcal{V}^t : S(v) = s\}$ denote the set of nodes of group $s$ present at step $t$. The probability that $v_{\text{new}}$ attaches to a node $v$ of the same group is proportional to:

$$w_+(v, \beta) = \deg_t(v) + (e^\beta - 1),$$

and to a node of a different group proportional to:

$$w_-(v) = \deg_t(v).$$

The probability that an edge created at step $t$ is intra-group is therefore:

$$p^t_{\text{intra}}(\beta) = \frac{\sum_{v \in \mathcal{V}^t_{S(v_{\text{new}})}} w_+(v, \beta)}{\sum_{v \in \mathcal{V}^t_{S(v_{\text{new}})}} w_+(v, \beta) + \sum_{v \notin \mathcal{V}^t_{S(v_{\text{new}})}} w_-(v)}.$$

Let $D^t_s = \sum_{v \in \mathcal{V}^t_s} \deg_t(v)$ denote the sum of degrees of group $s$ at step $t$, and $n^t_s = |\mathcal{V}^t_s|$. Setting $s = S(v_{\text{new}})$, this rewrites as:

$$p^t_{\text{intra}}(\beta) = \frac{D^t_s + n^t_s(e^\beta - 1)}{D^t_s + D^t_{\bar{s}} + n^t_s(e^\beta - 1)}.$$

The derivative with respect to $\beta$ is:

$$\frac{\partial p^t_{\text{intra}}}{\partial \beta} = \frac{n^t_s e^\beta \cdot D^t_{\bar{s}}}{\left(D^t_s + D^t_{\bar{s}} + n^t_s(e^\beta - 1)\right)^2},$$

where $\bar{s}$ denotes the opposite group. This quantity is strictly positive whenever $n^t_s > 0$ and $D^t_{\bar{s}} > 0$, conditions that hold for any non-trivial graph and any sufficiently large $t$. Hence $p^t_{\text{intra}}(\beta)$ is strictly increasing in $\beta$ at each growth step.

**Step 2: $\mathbb{E}[e_{00} + e_{11}]$ is a weighted average of $p^t_{\mathbf{intra}}(\beta)$.**  At each growth-step $t$, $m'_t$ edges are created, with probability $p^t_{\text{intra}}(\beta)$ of being intra-group. By definition, the total proportion of intra-group edges writes

$$e_{00} + e_{11} = \frac{\sum_t \sum_{k=1}^{m'_t} \mathbb{1}\{\text{edge } k \text{ at step } t \text{ is intra-group }\}}{\sum_t m'_t}$$

By linearity of expectation and the independence between $m'_t$ and the edge indicator, we obtain

$$\mathbb{E}[e_{00} + e_{11}] = \sum_t w_t \times p^t_{\text{intra}}(\beta), \qquad \text{with } w_t = \frac{\mathbb{E}[m'_t]}{\sum_t \mathbb{E}[m'_t]} > 0, \quad \sum_t w_t = 1 \tag{2}$$

Since each $p^t_{\text{intra}}(\beta)$ is strictly increasing in $\beta$ by step 1, so is $\mathbb{E}[e_{00} + e_{11}]$.

**Step 3: monoticity of Assort($\mathcal{G}$) in $p = e_{00} + e_{11}$.** We start by recalling that assortativity is defined as:

$$\text{ASSORT}(\mathcal{G}) = \frac{\text{Tr}(e) - \sum_{ij} e_{ij}^2}{1 - \sum_{ij} e_{ij}^2},$$

where $e_{ij}$ denotes the proportion of edges connecting group $i$ to group $j$. We assume that, conditionally on an edge being intra-group, the split between $e_{00}$ and $e_{11}$ is independent of $\beta$ and determined by $\alpha$ alone.

$$e_{00} = \alpha p, \quad e_{11} = (1 - \alpha)p, \quad e_{01} = e_{10} = \frac{1 - p}{2}.$$

Under this assumption, setting $\kappa = \alpha^2 + (1 - \alpha)^2$, we obtain

$$\sum_{i,j} e_{ij}^2 = \kappa p^2 + \frac{(1 - p)^2}{2}.$$

Writing Assort $= N/D$ with $N(p) = p - \kappa p^2 - \frac{(1-p)^2}{2}$ and $D(p) = 1 - \kappa p^2 - \frac{(1-p)^2}{2}$, and noting that $N = D - (1 - p)$:

$$\frac{d}{dp}\text{Assort} = \frac{N'D - ND'}{D^2} = \frac{D + (1-p)D'}{D^2} \tag{3}$$

Expanding explicitly:

$$D + (1 - p)D' = 1 - \kappa p^2 - \frac{(1 - p)^2}{2} + (1 - p)\big(1 - p(1 + 2\kappa)\big)$$

$$= (1 - \kappa) + (1 - p)^2 \left(\kappa + \frac{1}{2}\right) \tag{4}$$

This quantity is strictly positive for all $p \in [0, 1]$ and $\alpha \in (0, 1)$. Therefore $\frac{d}{dp}\text{Assort} > 0$ on $[0, 1]$.

Since $\mathbb{E}[e_{00} + e_{11}]$ is strictly increasing in $\beta$ (Steps 1–2) and Assort is strictly increasing in $e_{00} + e_{11}$ (Step 3), $\mathbb{E}[\text{Assort}(G)]$ is strictly increasing in $\beta$.

$\square$

**Remark 3.** The argument above covers the case without anchor nodes. When the anchor mechanism is active, the homophilic bias operates through the selection of the anchor node $u$ itself via $P_{\beta,S}$, which preserves the monotonicity in $\beta$ by the same reasoning applied to the first attachment step.

## B.2 Dataset Description

We provide a more detailed description of the three scenarios to which we compare the generated graphs.

**Polblogs (*Opinion*)** It consists of networks of opinion blogs, where the sensitive attribute is defined as the political affiliation of bloggers (left wing and right wing). LP here refers to blog recommendations to bloggers and raises challenges with fairness, such as filter bubbles constraining users within narrow opinion spectrum.

**Facebook (*Friendship*)** It is an ego network of friendship among Facebook members, with the sensitive attribute being the users' gender. Common LP tasks in these types of networks include recommending friends or professional connections. The natural tendency for homophily in social relationships can exacerbate existing societal biases, leading to the amplification of discriminatory outcomes, such as the perpetuation of gender or racial biases in professional opportunities, or the segregation of communities within the network.

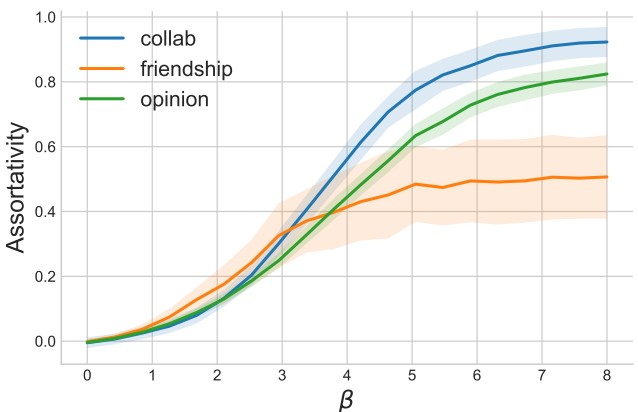

| | PC1 | PC2 |
|---|---|---|
| CLOSENESS | 0.353 | -0.114 |
| BETWEENNESS | 0.105 | -0.166 |
| PRESTIGE | 0.315 | -0.087 |
| DEGREE | 0.365 | -0.043 |
| CONSTRAINT | 0.384 | -0.040 |
| DENSITY | 0.363 | 0.044 |
| HETEROGENEITY | -0.288 | -0.223 |
| ISOLATION | 0.234 | 0.374 |
| DIAMETER | 0.209 | 0.408 |
| CONTROL | 0.103 | -0.301 |
| ASSORTATIVITY | 0.057 | 0.469 |
| AVG MIXED DIST | -0.102 | 0.476 |
| POWER EXP | -0.238 | 0.082 |
| INFO UNFAIRNESS | 0.300 | -0.216 |

Figure 6: Effect of $\beta$ parameter on ASSORTATIVITY in the three generation processes. Here, $\alpha$ parameter has been set to 0.5 and 100 different seeds have been used to form the standard-deviation error bands.

Table 3: Loadings of the first two principal components for the PCA of structural bias measures over the generated corpus.

**Collab (*Collab*)** The network dataset represents scientific collaborations among researchers on the HAL platform, from 2017 to 2022, where edges are co-authorship in research papers and the sensitive attribute is the researcher's gender (name-based association, according to INSEE). Note that sensitive attributes in these networks could also include factors such as nationality, or institutional affiliation. The LP task involves suggesting potential collaborators or recommending relevant content, which is susceptible to fairness concerns, as biases in collaboration suggestions can result in the marginalization of underrepresented groups, reinforcing existing disparities in access to resources and opportunities.

### B.3   Additional results

Figure 6 illustrates the effect of the homophily parameter $\beta$ on assortativity for a balanced group configuration ($\alpha = 0.5$) across the three use cases. The resulting curves are monotonically increasing, confirming that $\beta$ exerts the intended directional control over assortativity. A saturation plateau is also observed beyond a certain threshold, beyond which assortativity reaches its maximum value, reflecting structural differences in the connectivity patterns induced by the three variants of the generative algorithm.

Table 4 compares real-world graphs with their synthetic counterparts obtained by fitting the generative parameters to each dataset. Since the number of nodes and class imbalance are directly encoded in the algorithm, exact replication of these quantities is guaranteed by construction. Assortativity values are closely matched across all use cases, demonstrating the relevance of the homophily parameter $\beta$ as a control variable. Degree-related measures — average degree and density — are also well reproduced in all three settings, reflecting the effectiveness of the Gamma-distributed degree parameterization in controlling graph connectivity.

## C   Structural Bias Heatmaps

Next, we present the evolution of the different structural bias measures across sensitive class imbalance $\alpha$ and homophily parameter $\beta$, in the three considered use cases. We observe that the different biases evolve according to distinct patterns, for example being more sensitive to $\beta$, like ASSORTATIVITY, or to $\alpha$, like HETEROGENEITY. Moreover, the variation profiles can differ greatly depending on the use case, as for BETWEENNESS, confirming the diversity of the topologies involved in the three use cases.

| | *Opinion* | | *Friendship* | | *Collab* | |
|---|---|---|---|---|---|---|
| | Real | Synth. | Real | Synth. | Real | Synth. |
| Nodes | 1222 | 1222 | 1034 | 1034 | 860 | 860 |
| Class imbalance | 0.52 | 0.52 | 0.66 | 0.66 | 0.53 | 0.53 |
| ASSORT. | 0.81 | 0.82 | 0.06 | 0.06 | 0.10 | 0.08 |
| Mean degree | 27.4 | 28.6 | 52 | 33 | 6.1 | 6.5 |
| $\mathcal{V}_0$ degree | 27.6 | 29.3 | 53 | 33 | 6.5 | 6.8 |
| $\mathcal{V}_1$ degree | 27.1 | 27.8 | 51 | 33 | 5.7 | 6.2 |
| Graph Density | 0.02 | 0.02 | 0.05 | 0.03 | 0.007 | 0.008 |

Table 4: Structural properties of real and fitted synthetic graphs across the three use cases.

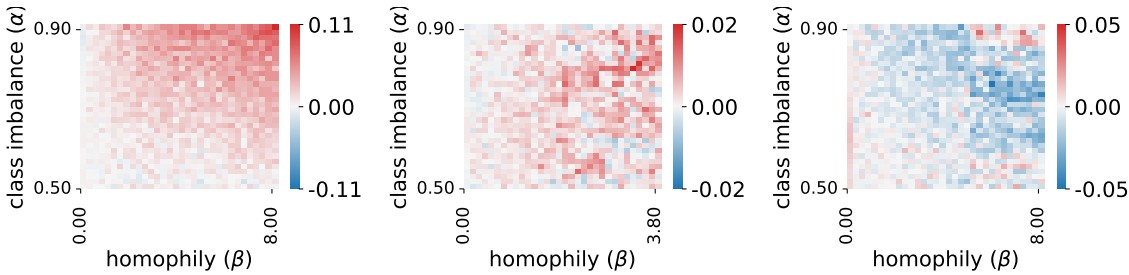

Figure 7: CLOSENESS values in *Opinion* (left), *Friendship* (middle), and *Collab.* (right) use cases.

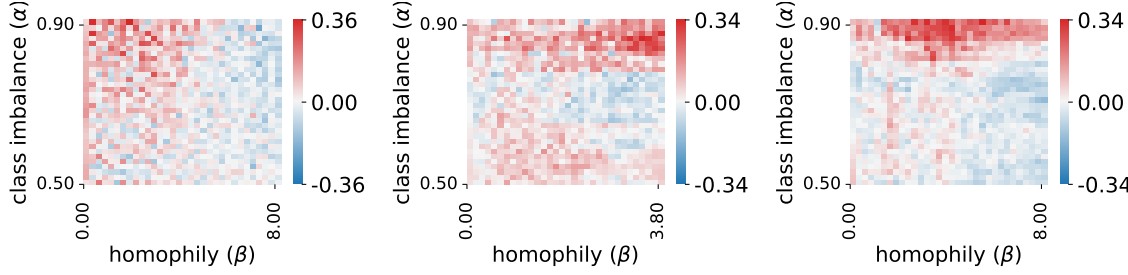

Figure 8: BETWEENNESS values in *Opinion* (left), *Friendship* (middle), and *Collab.* (right) use cases.

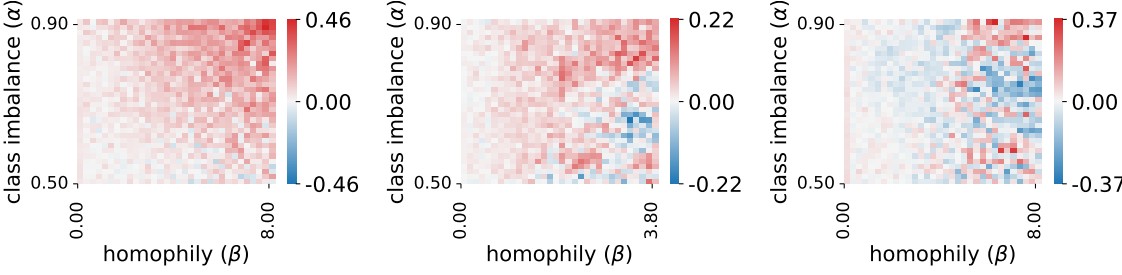

Figure 9: PRESTIGE values in *Opinion* (left), *Friendship* (middle), and *Collab.* (right) use cases.

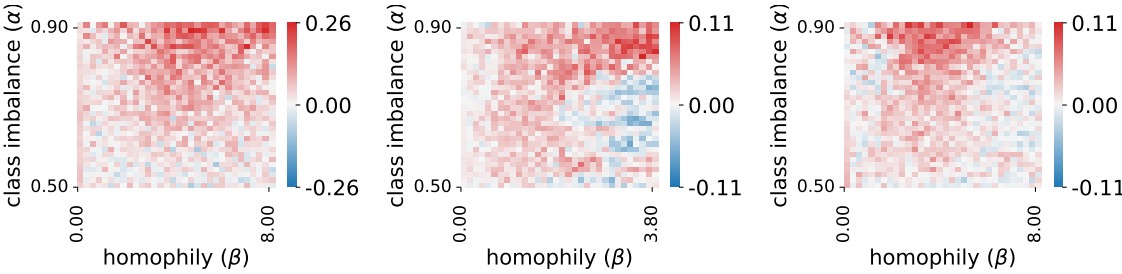

Figure 10: DEGREE values in *Opinion* (left), *Friendship* (middle), and *Collab.* (right) use cases.

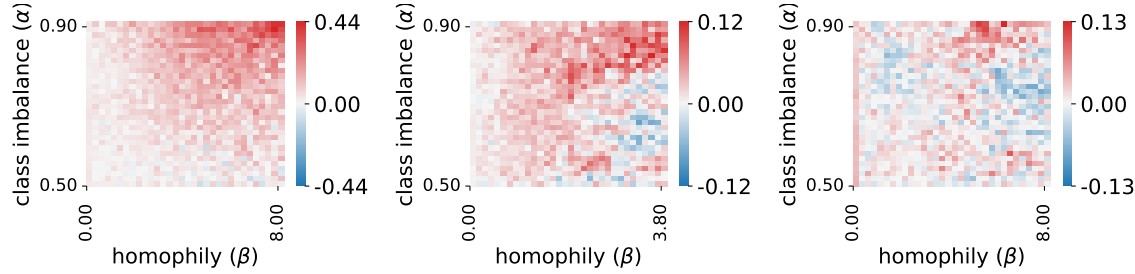

Figure 11: CONSTRAINT values in *Opinion* (left), *Friendship* (middle), and *Collab.* (right) use cases.

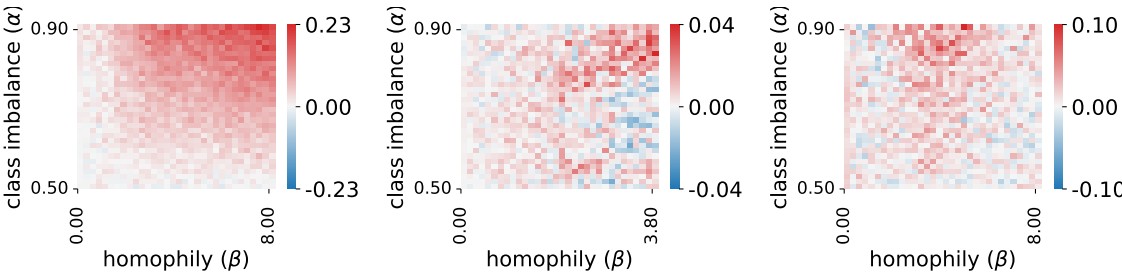

Figure 12: DENSITY values in *Opinion* (left), *Friendship* (middle), and *Collab.* (right) use cases.

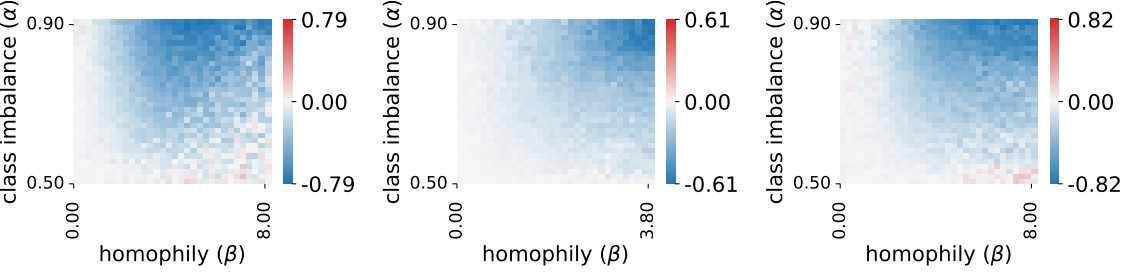

Figure 13: HETEROGENEITY values in *Opinion* (left), *Friendship* (middle), and *Collab.* (right) use cases.

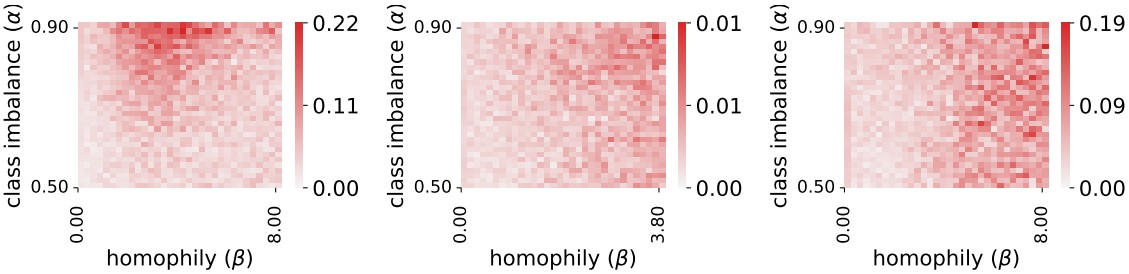

Figure 14: ISOLATION values in *Opinion* (left), *Friendship* (middle), and *Collab.* (right) use cases.

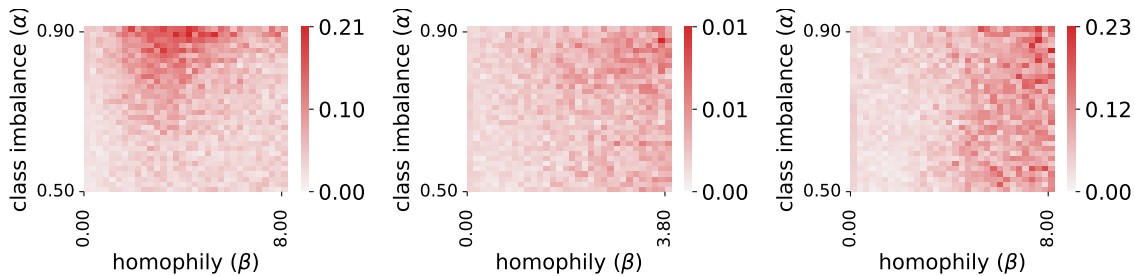

Figure 15: DIAMETER values in *Opinion* (left), *Friendship* (middle), and *Collab.* (right) use cases.

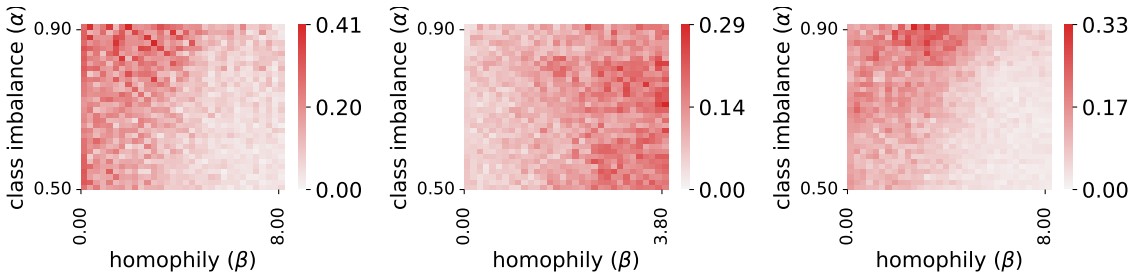

Figure 16: CONTROL in *Opinion* (left), *Friendship* (middle), and *Collab.* (right) use cases.

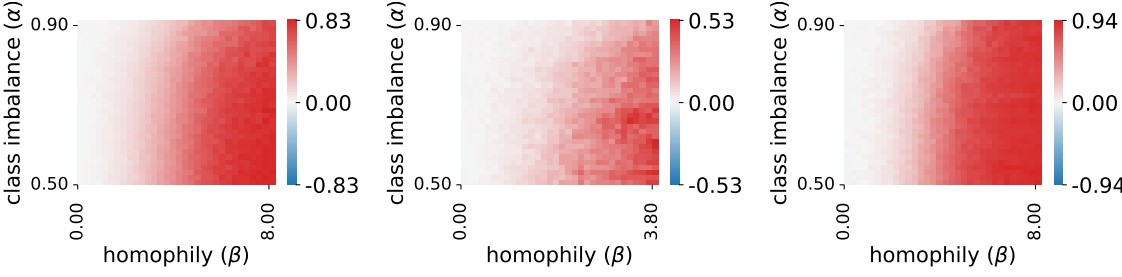

Figure 17: ASSORTATIVITY in *Opinion* (left), *Friendship* (middle), and *Collab.* (right) use cases.

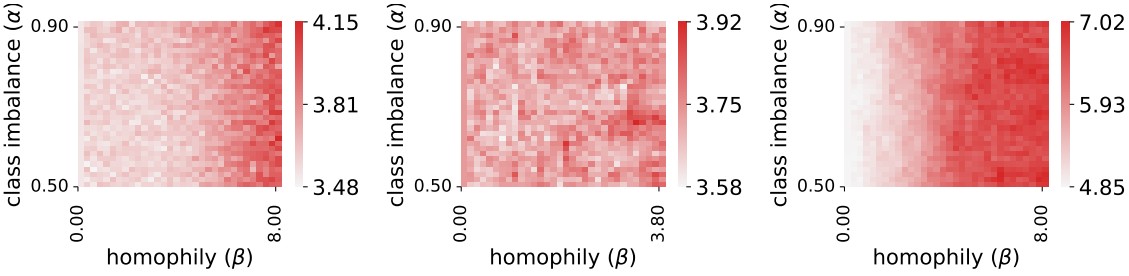

Figure 18: AVG MIXED DIST in *Opinion* (left), *Friendship* (middle), and *Collab.* (right) use cases.

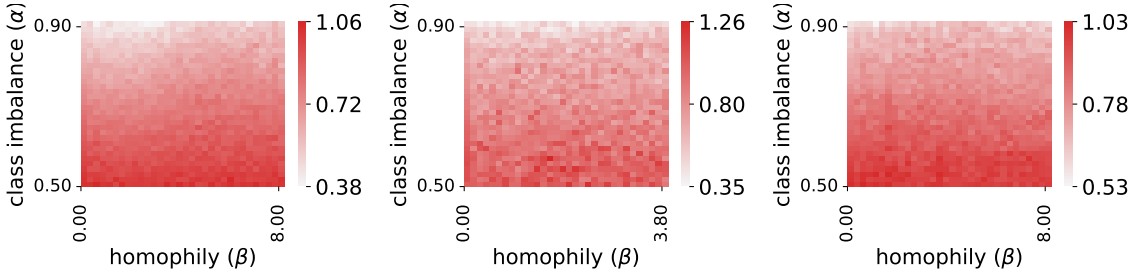

Figure 19: POWER EXP in *Opinion* (left), *Friendship* (middle), and *Collab.* (right) use cases.

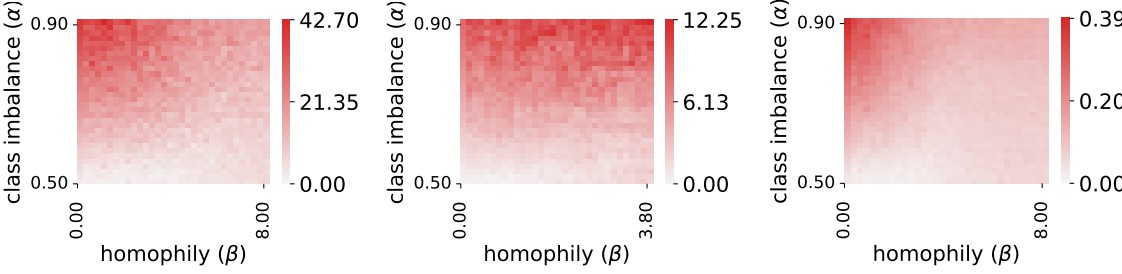

Figure 20: INFO UNFAIRNESS in *Opinion* (left), *Friendship* (middle), and *Collab.* (right) use cases.

## C.1 Correlation heatmaps

Here, we present the correlation matrices between the structural bias measures across the three scenarios. Only significant correlations (p-value < 0.01) are displayed in color.

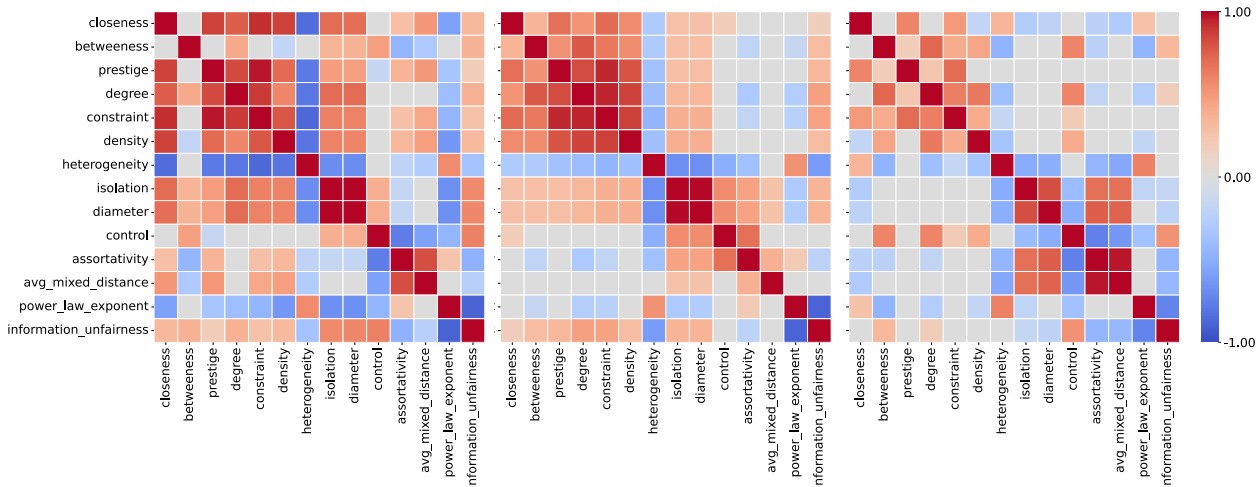

Figure 21: Correlations between structural bias measures in *Opinion* (left), *Friendship* (middle), and *Collab.* (right) use cases.

Figure 21 shows that the correlations between bias measures can vary considerably across scenarios, such as assortativity alternating between positive and negative correlations with control. However, some correlations remain consistent, notably between degree and constraint, both of which quantify node connectivity at different scales. This consistency underlines the existence of structural properties common to all scenarios.

## D   Predictive Metrics Heatmaps

As in Appendix C, we present here the evolution of predictive metrics (fairness and performance ones) across sensitive class imbalance $\alpha$ and homophily parameter $\beta$, in the three considered use cases. In the following, each subsection refers to a particular LP model.

Overall, the models involved demonstrate strong performance, confirming the relevance of our results regarding the fairness metrics. We also observe that, as expected, performance generally increases with homophily, and that unfairness grows with both our homophily and sensitive class imbalance parameters $\beta$ and $\alpha$. This highlights the relevance of these two parameters as primary factors driving predictive disparities.

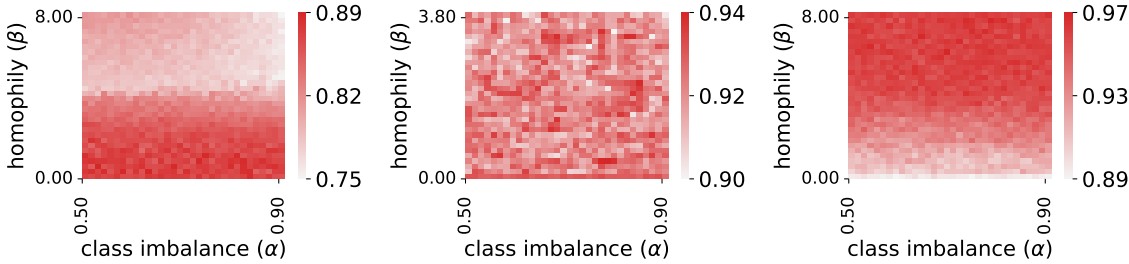

Figure 22: *AUC* for **GCN** models in *Opinion* (left), *Friendship* (middle), and *Collab.* (right) use cases.

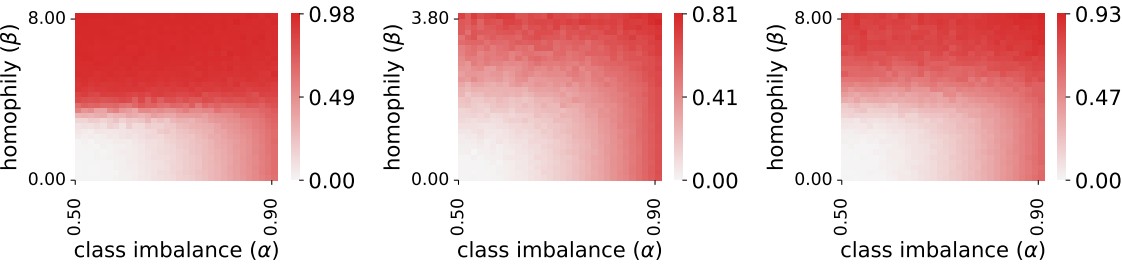

Figure 23: *SP*@10 for **GCN** models in *Opinion* (left), *Friendship* (middle), and *Collab.* (right) use cases.

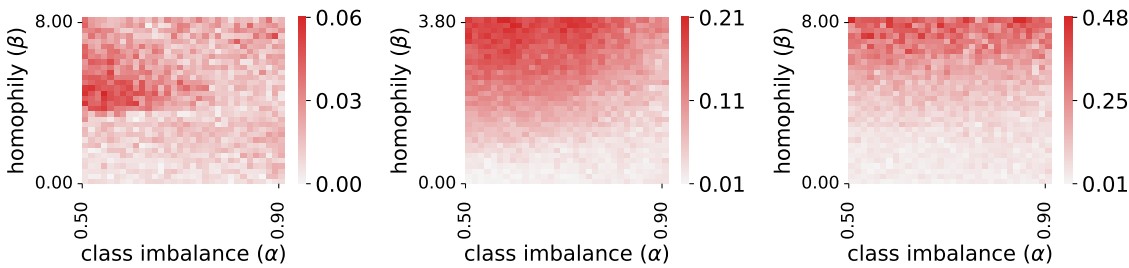

Figure 24: *EO*@10 for **GCN** models in *Opinion* (left), *Friendship* (middle), and *Collab.* (right) use cases.

# E   Importance plots

## E.1   Graph Convolutional Network (GCN)

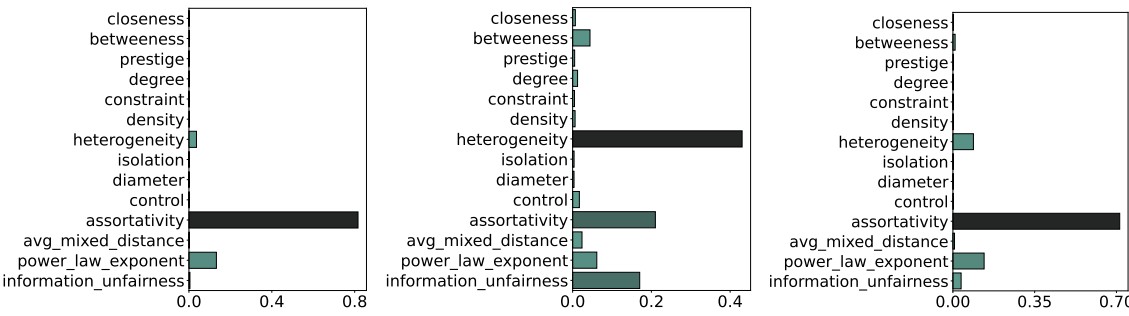

Figure 25: Feature importance scores from the structural bias regression for $SP$ metric and $GCN$ model in *Opinion* (left), *Friendship* (middle), and *Collab.* (right) use cases.

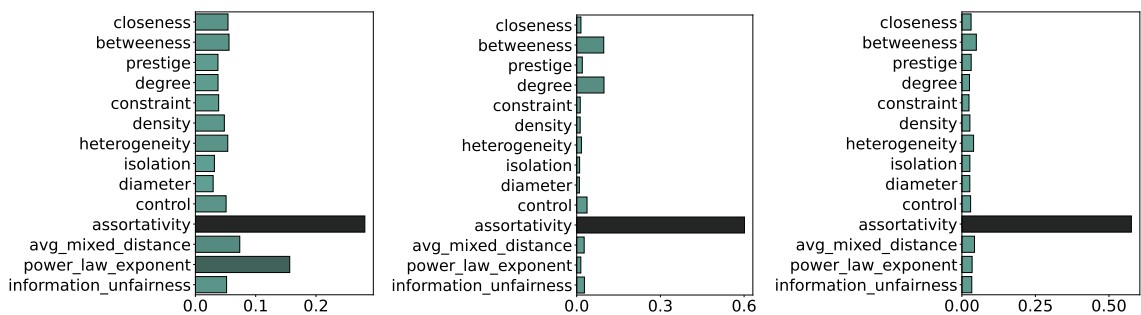

Figure 26: Feature importance scores from the structural bias regression for *EO* metric and *GCN* model in *Opinion* (left), *Friendship* (middle), and *Collab.* (right) use cases.

## E.2 Node2Vec (N2V)

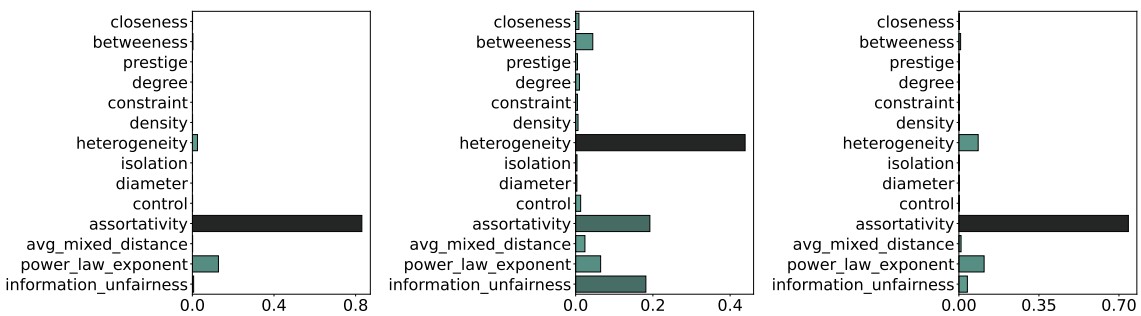

Figure 27: Feature importance scores from the structural bias regression for *SP* metric and *N2V* model in *Opinion* (left), *Friendship* (middle), and *Collab.* (right) use cases.

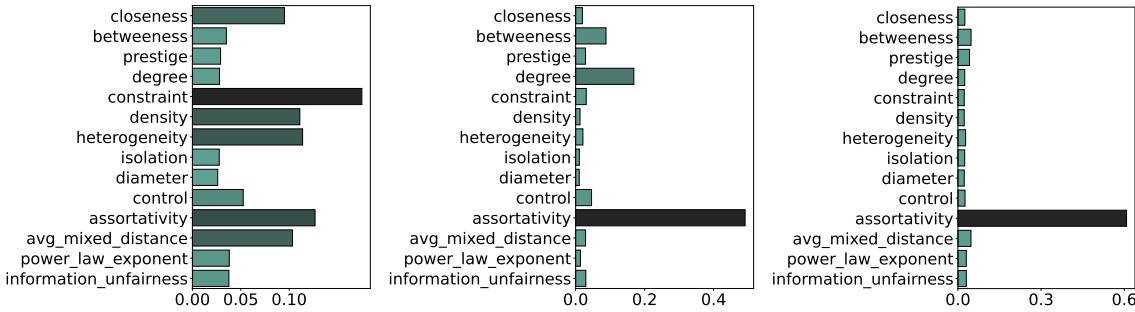

Figure 28: Feature importance scores from the structural bias regression for *EO* metric and *N2V* model in *Opinion* (left), *Friendship* (middle), and *Collab.* (right) use cases.

### E.3 Singular Value Decomposition (SVD)

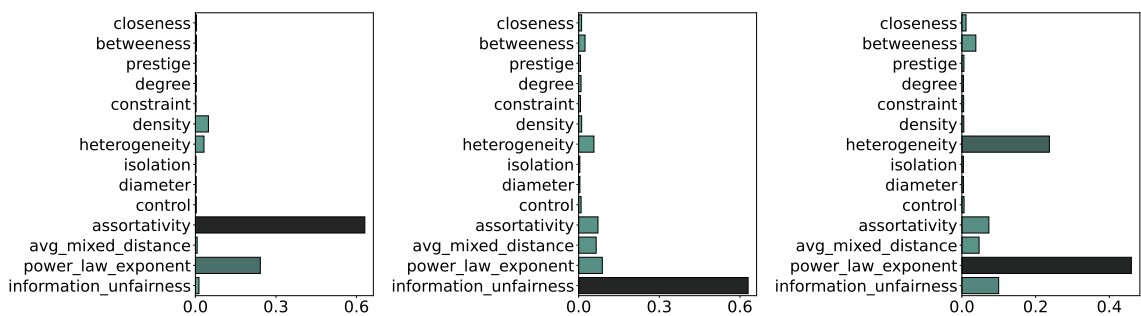

Figure 29: Feature importance scores from the structural bias regression for $SP$ metric and $SVD$ model in *Opinion* (left), *Friendship* (middle), and *Collab.* (right) use cases.

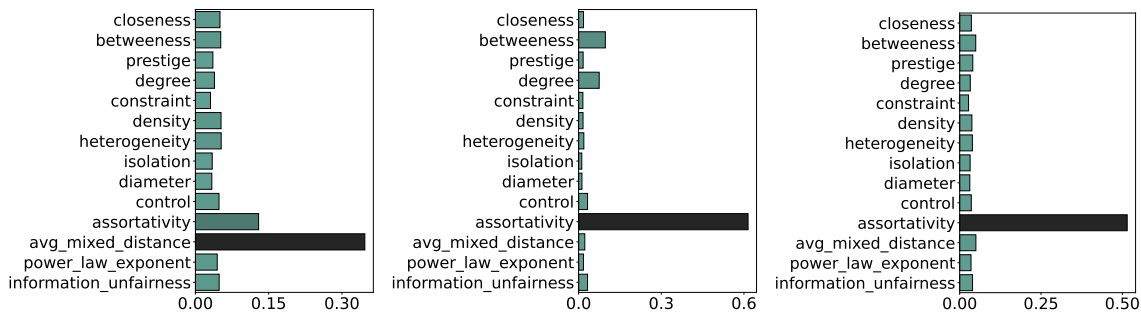

Figure 30: Feature importance scores from the structural bias regression for $EO$ metric and $SVD$ model in *Opinion* (left), *Friendship* (middle), and *Collab.* (right) use cases.

## F  Additional results for H2

## G  Additional results for H3

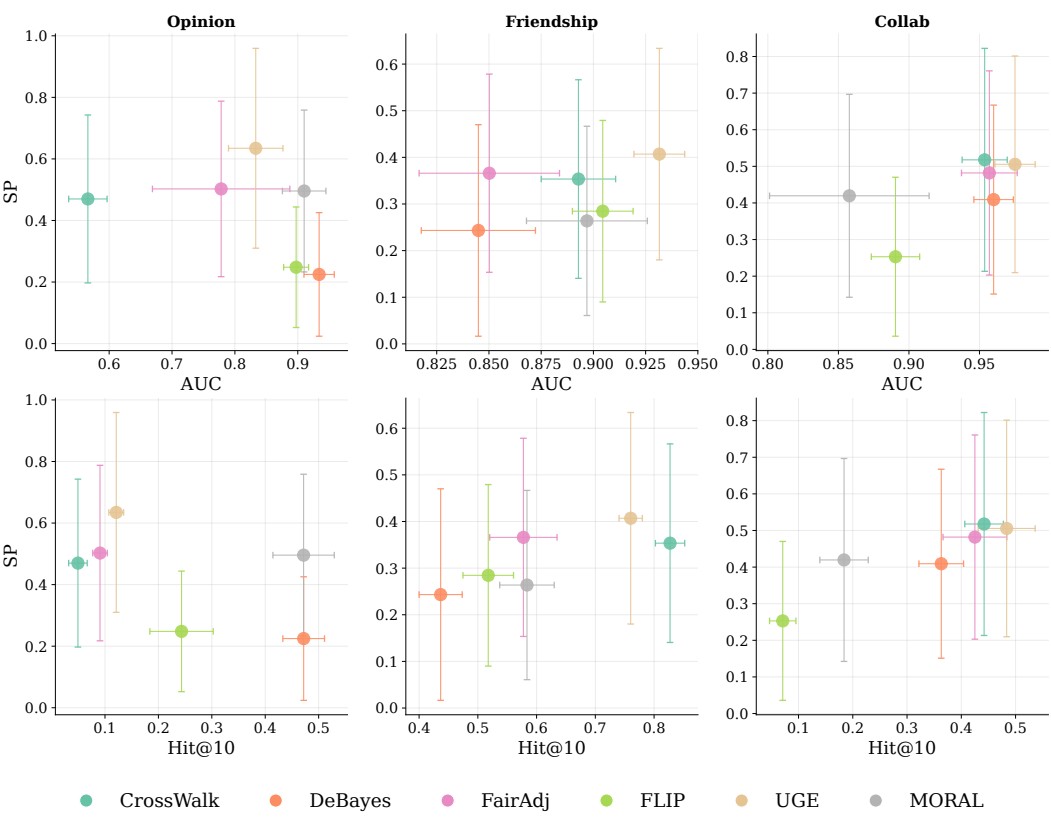

Figure 31: Distribution of fairness and performance outcomes across the synthetic graph corpus, for each fair LP model and use case. Each point represents the mean over all synthetic graphs; error bars show ± standard deviation.

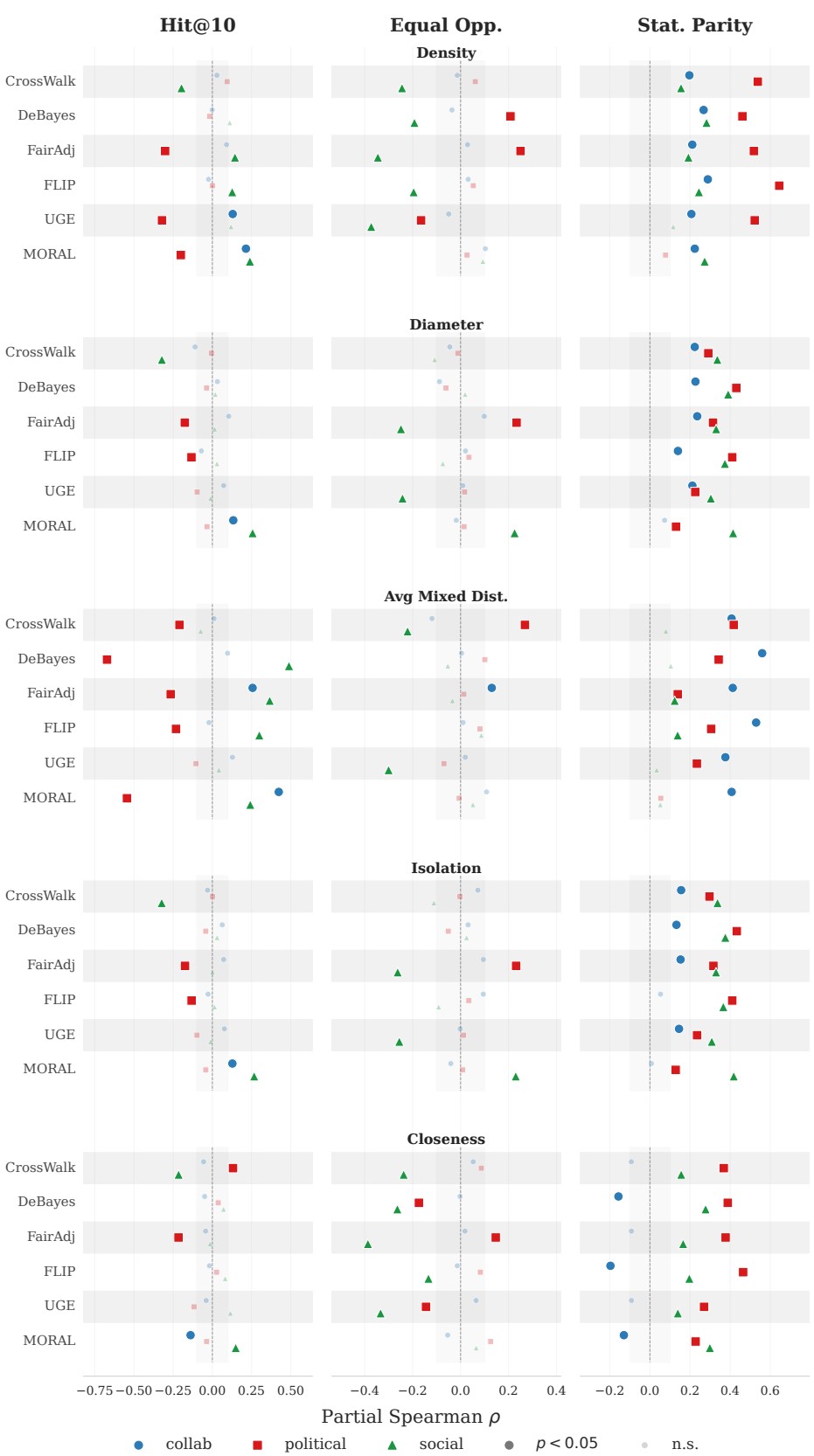

Figure 32: Partial Spearman $\rho$ between the structural biases and fairness/performance metrics across models and use cases. Marker opacity encodes statistical significance ($p < 0.05$).

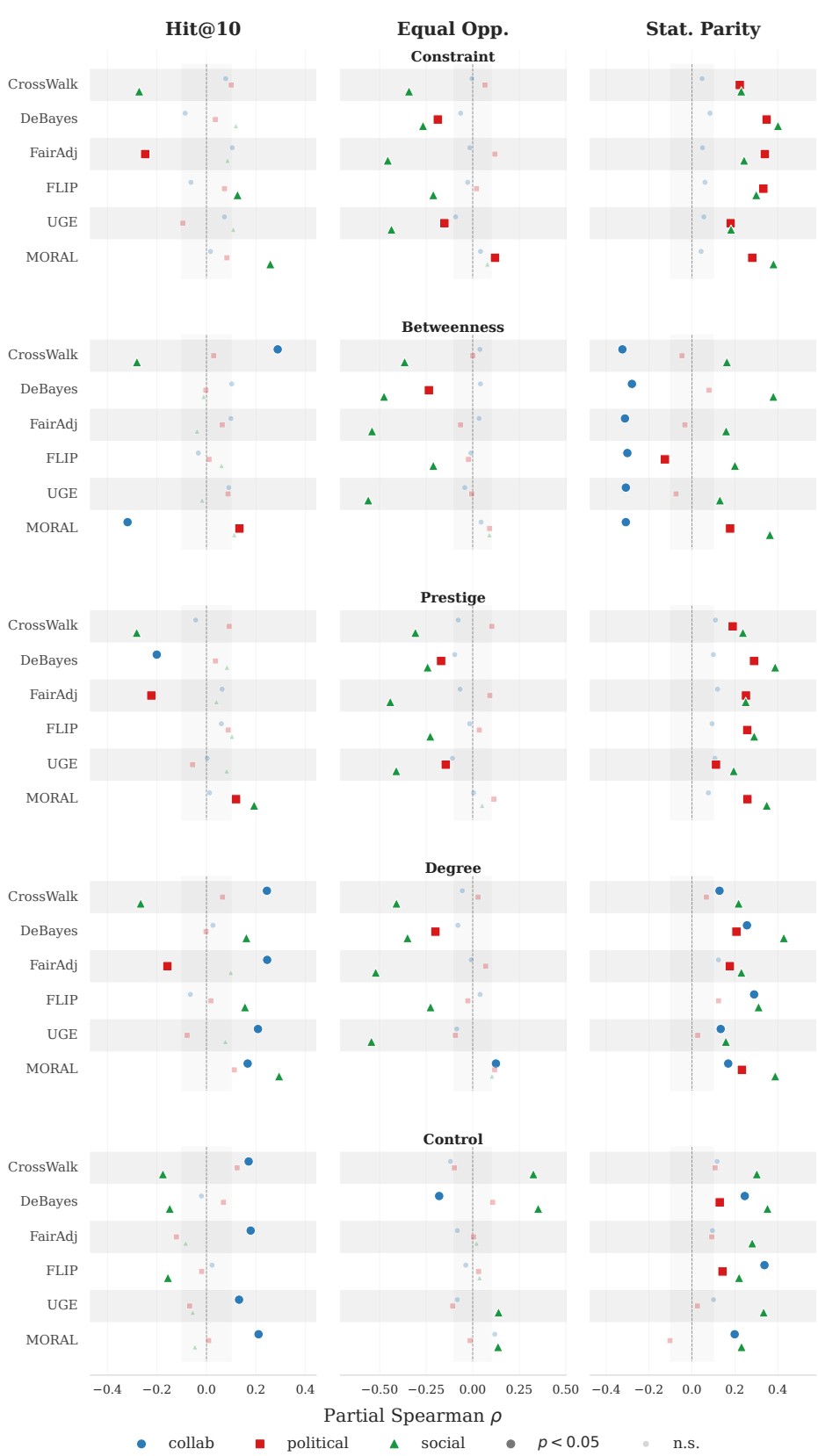

Figure 33: Partial Spearman $\rho$ between the structural biases and fairness/performance metrics across models and use cases. Marker opacity encodes statistical significance ($p < 0.05$).

