# OpenReview forum: "Structural Bias Beyond Homophily: A Study of Fairness in Link Prediction"
_TMLR — Under review for TMLR_

### Review · Reviewer_pkZY · 2026-07-05

**Summary Of Contributions:**

This paper poses that the fairness of link-prediction (LP) methods is driven by graph topology, which goes _beyond_ homophily. The authors contribute (1) a taxonomy of structural bias measures spanning node- and group-level, topological and flow-based properties, (2) a modular extension of the Barabási–Albert generator that lets them control class imbalance ($\alpha$) and a homophily parameter ($\beta$) while calibrating to three real-world use cases (_Opinion_, _Friendship_, _Collab_), and (3) an empirical study testing three hypotheses. H1: classical LP fairness is determined by structural bias, H2: fair methods only help when they target the right source of bias, H3: biases beyond homophily still matter once assortativity is held fixed.

**Audience:**

Yes

**Audience Explanation:**

Yes at a small scale. Folks working on fair graph learning / structural bias will find the taxonomy and controlled generator useful. The "assortativity dominates but isn't exclusive" observation is worth knowing. The audience is niche given the diagnostic-only scope and three use cases.

**Claims And Evidence:**

Yes

**Claims Explanation:**

- Partially, and I think that most of the gaps can be closed by adjusting claims rather than needing a significant amount of new experimentation.
- The main "beyond homophily" framing is where the claims and supporting evidence drift apart somewhat. Section 3.1.2 defines assortativity *as* homophily, but H3 then hold assortativity fixed and calls the residuals "beyond homophily". Attribute assortativity is only one type of homophily, and unfortunately there's no single single agreed upon measure [1, 2]. It would be best to be explicit that assortativity is the homophily proxy being controlled (so it's more like "beyond attribute assortativity" vs. "beyond homophily" but this framing would be a significant change).
- I'd separate two claims in H1. "Topology predicts fairness" ($R^2 \gt 0.90$) is reasonably robust. "*Which* bias dominates" leans on the impurity-based importances from a single Random Forest regressor, which is known to be unstable and biased toward correlated features [3] (Figure 21 shows these measure can be heavily correlated). I'd suggest adding permutation importance + a significance test (a block permutation p-value, since there's no analytic one), and ideally a second regressor (XGBoost or even a linear model). The nice finding here at least is that assortativity dominates across most dataset $\times$ method pairs.
- H2 is, to me, the most interesting result and slightly undersold. The claim that fair methods help by targeting the _right_ source of bias rather than by intervention strength is interesting and the per-mechanism reasoning (DeBayes' degree correction, MORAL's separate encoder, FLIP's adversarial loss) is experimentally sound. That said, these experimental results are fitted to the outcomes in Figure 4 rather than tested, so the causal framing outruns the evidence. Either soften to interpretation ("consistent with," "we conjecture") or add a small ablation that disables each mechanism and checks whether the fairness advantage disappears (e.g., for DeBayes, remove/neutralize the degree correction and check whether inter-hub performance in _Opinion_ collapses; for MORAL, merge the separate intergroup encoder back into one and see if the gain vanishes. If the mechanism is really the cause, disabling it should erase the effect.)

**Requested Changes:**

### Major:
- Should disentangle homophily from assortativity. State explicitly which homophily measure the claim rests on, justify assortativity as the control variable, and show the "beyond homophily" residual isn't just unmodeled homophily. Engaging with homophily-measure literature [1, 2] and some fairness/assortativity links [4] would sharpen this. Since this is the core claim it should be airtight.
- Strengthen H1 statistically by adding a permutation/block-permutation significance test for the $R^2$ and importance results, and validate with $\geq 1$ additional regressor. One RF is not enough evidence that "topology determines fairness."
- The conclusion should be scoped to the graph sizes actually tested, or add larger-graph evidence. A limitation block would be find if new large-scale runs aren't feasible.

### Minor:
- Section 3.11 seems to mix up node- and group-level language. The "quantify bias… across sensitive groups" / "advantage for the non-sensitive group" phrasing appears before group-level measures are introduced in 3.1.2. If the intent is "node-level measures aggregated to a group summary," say so and contrast it clearly with 3.1.2; otherwise the "group" references read as misplaced.
- Assortativity should be motivated a bit more in Section 3. Currently, in this section, it reads as just 1 of 4 group measures in Table 1, yet it's the pivot for every downstream test. Flag its special role up front so the reader isn't surprised by its dominance in Section 4.
- Explain why the calibrated parameters differ so much across use cases in Section 3.3. Appendix B.2 describes the settings well, but a few sentences connecting each dataset's objective to its resulting structural properties (why _Opinion_ lands at assort. 0.82 vs. _Friendship_ at 0.06) would make the calibration understandable.
- Broaden beyond the three scenarios, or justify why these three are fully representative of fairness across all graph settings.
- There are additional topological descriptors which could be explored. The current taxonomy is centrality/flow-based, however there is a good amount of multi-scale descriptors from persistent homology which have been sued to characterize LP-relevant graph structure [5, 6] and might capture "structure beyond homophily" more directly than the current measures.

**References:** \
[1] Platonov et al., "Characterizing Graph Datasets for Node Classification: Homophily–Heterophily Dichotomy and Beyond" _NeurIPS_, 2023 \
[2] Mironov et al., "Revisiting Graph Homophily Measures" _LoG_, 2025 \
[3] Strobl et al., "Bias in random forest variable importance measures: Illustrations, sources and a solution" _BMC Bioinformatics_, 2007 \
[4] Loveland et al., "On Graph Neural Network Fairness in the Presence of Heterophilous Neighborhoods" _KDD DLG Workshop_, 2022 \
[5] Yan et al., "Link Prediction with Persistent Homology: An Interactive View" _ICML_, 2021 \
[6] Bhatia et al., "Understanding and Predicting Links in Graphs: A Persistent Homology Perspective" _ArXiv_, 2018

---

### Review · Reviewer_mUJq · 2026-07-06

**Summary Of Contributions:**

The paper studies fairness in graph link prediction and argues that existing work over focuses on homophily. The paper has three main contributions. First, it organizes structural bias measures beyond homophily into a taxonomy. Second, it proposes a controllable BA-style synthetic graph generator with parameters for sensitive-group imbalance and homophily. Third, it uses the generated graphs to show that both classical and fairness-aware link prediction methods remain sensitive to structural biases even when assortativity is controlled. The high-level question is important and interesting, therefore, the paper has a useful diagnostic direction. However, the current version has substantial problems in novelty positioning, mathematical correctness, causal and statistical interpretation, experimental validation, and clarity of evaluation protocol.

**Audience:**

Yes

**Audience Explanation:**

I think researchers working on graph representation learning, fair graph learning, link prediction, recommender systems, and benchmark design would be interested in the paper's finding. The paper addresses a relevant problem: fairness in graph link prediction for socially consequential settings such as job recommendation, friendship recommendation, and collaboration recommendation. Its central message, that fairness failures should not be attributed to homophily alone, is potentially useful for this audience. The taxonomy of structural biases and the synthetic testing framework could help researchers evaluate fair link prediction methods under a wider range of graph topologies.

**Broader Impact Concerns:**

The paper has a Broader Impact Statement section. I suggest the authors include the following two discussions in this section.

1. Regarding the Collab dataset, where gender is inferred using name based association. The paper should discuss consent, coverage, ambiguity handling, nonbinary identities, cultural bias, or error rates in gender inference. Since the sensitive attribute is central to the structural bias measures and fairness metrics, label noise or binary gender assumptions could materially affect the conclusions.

2. The paper also relies mainly on Statistical Parity and Equal Opportunity, but these metrics may not capture all relevant harms in the studied applications. For example, in friendship recommendation, user preference and consent are also important. The Broader Impact Statement should more clearly discuss the normative limits of these metrics.

**Claims And Evidence:**

No

**Claims Explanation:**

1. The paper’s high-level framing is useful, but many of its core ingredients are not new. Dyadic fairness for link prediction is already introduced as the basis of the paper itself through Li et al. 2021, and many structural measures in the taxonomy are classical network measures or previously proposed fairness-oriented graph measures. The paper’s real contribution is closer to a benchmark and diagnostic framework, not a new fairness theory or a new class of structural-bias metrics.

2. The paper does not sufficiently justify its choice of fairness metrics (SP and EO). SP may be inappropriate when edge base rates differ, while EO conditions on observed test edges that may already encode historical structural bias. For example, in social graphs, observed edges often reflect unequal opportunity, platform exposure, segregation, and previous recommendation effects. Therefore EO may reward a model for reproducing historically biased structure.

3. Related to comment 2: Why alternative notions such as counterfactual fairness, exposure fairness, or node-level opportunity fairness are not considered. This is especially important because the paper makes claims about structural sources of unfairness, for which causal or counterfactual fairness notions would be natural alternatives.

4. The paper discusses FairAdj, FairGen, GenCAT, biased preferential attachment, and related homophily work, but it does not sufficiently position itself against very close work on structural bias and fair graph generation. FairWire [1] is especially relevant because it studies structural bias causing disparity in dyadic relation prediction and proposes a fair graph generation framework. That is very close to this paper’s combination of dyadic fairness, graph generation, and structural bias analysis.

5. The definition of assortativity in A.2 and that used in step 3 of B.1 are inconsistent. Both are not the standard nominal assortativity, which is usually expressed using the mixing matrix and row/column marginals. Please clarify the definition used in this paper.

6. Proposition 2 claims that for fixed $\alpha$ and graph size, expected assortativity increases monotonically with $\beta$. The proof assumes a clean relationship between the probability of intra-group edge formation and global assortativity. However, the actual generator includes anchor nodes and, in the Friendship setting, uses $m'=0.55 deg(u)+3$, where $u$ is the anchor node. Since the anchor is selected through a homophily biased process, $m'$ may depend on $\beta$, sensitive group, and degree. This undermines the independence assumptions used in the proof.

7 Related to comment 6: Figure 6 only shows the effect of $\beta$ on on assortativity for $\alpha=0.5$. The paper claims that $\beta$ can be used as a structural intervention on assortativity across the generated corpus. The empirical check should cover the full range of $\alpha$, all generator variants, and the anchor-based mechanisms.

8. The paper reports many partial correlations across structural biases, metrics, models, and use cases. Figures 5, 32, and 33 involve many hypothesis tests and mark significance using $p<0.05$. Without false discovery rate or family wise error correction, some significant results may be false positives.

9. The paper relies heavily on synthetic graphs. Therefore, generator validation is central. But the validation is weak. Table 4 shows that the Friendship synthetic graph does not match the real graph well: the real mean degree is 52, while the synthetic mean degree is 33; the real density is 0.05, while the synthetic density is 0.03. This is a major mismatch because degree and density strongly affect link prediction and fairness outcomes. Also, I feel this validation uses too few properties, I think the paper needs much stronger evidence that these graphs realistically capture the structural regimes relevant to fairness.

10. For H1 random forest result, the result reports high $R^2$ when predicting SP and EO from structural bias measures, often near 1.0. This may be partly tautological. SP and EO are based on intra-group versus inter-group prediction behavior. Many structural features used as inputs, such as assortativity, heterogeneity, mixed distance, and information unfairness, also encode intra-group/inter-group graph mixing. So it is not surprising that these features predict the fairness metrics. I would suggest the authors show how much $R^2$ is explained by $\alpha$, $\beta$, group imbalance, and assortativity alone by some simple baselines.

11. The H2 evidence is mostly descriptive. The paper needs targeted analyses showing that each method’s behavior is actually driven by the claimed structural bias source.

[1] Kose, O. D., & Shen, Y. (2024). FairWire: Fair graph generation. Advances in neural information processing systems, 37, 124451-124478.

**Requested Changes:**

**Proposed revisions critical to securing a recommendation for acceptance**

I would suggest revisions based on my previous comments.

**Proposed adjustments that would strengthen the work**

1. Why a Gamma distribution for $m'$ is sufficient? This choice needs more justification and discussion. Also, note degrees must be integers, please clarify how you handle it.

1. SP@10 and EO@10 should be formally defined, since they are not the exact SP and EO definitions. Similarly, please clarify HitRank and Hit@10.

2. In algorithm 1: In the anchor setting, the algorithm samples  $m'$ neighbors from the anchor ego-network and then adds the anchor node itself. Unless the anchor can already be included in the sampled neighbors, the new node receives $m'+1$ edges rather than $m'$.

3. The work reports mainly $R^2, feature importances, and partial correlations, but it does not consistently provide confidence intervals or robustness intervals. Since the main conclusions rely heavily on statistical associations, uncertainty estimates are necessary.

4. The H3 section says “The other biases are in Appendix ??”. Appendix B.1 should be titled “Proof of Proposition 2.” instead of “Proof of Proposition 1.”.

5. Section F is missing.

6. The appendix defines "constraint" as the sum of neighbors’ degrees. This is not a stand constraint measure, maybe it is better to rename it.

---

### Review · Reviewer_22fK · 2026-07-20

**Summary Of Contributions:**

**Summary**: This oaoer challenges the common practice of exclusively associating structural bias in graph link prediction (LP) with homophily. To investigate the broader topological factors driving algorithmic unfairness, the authors introduce a comprehensive taxonomy of structural biases that categorizes both node-level and group-level metrics, encompassing topological as well as flow-based properties. Furthermore, they develop a parametrizable graph generation process extending the Barabási-Albert model, which allows them to synthesize and evaluate graphs with highly controlled structural properties calibrated to real-world datasets. Through extensive empirical evaluations across both classical and fairness-aware LP models, the authors demonstrate that fairness outcomes are heavily dependent on the underlying graph topology. Ultimately, their findings reveal that current fairness interventions remain highly sensitive to structural biases beyond homophily—such as heterogeneity, power-law exponent ratio, and information unfairness—even when homophily is held constant, highlighting the need for more structurally grounded evaluation practices in fair graph learning.


**Strengths**

1. The authors present a well-founded argument for evaluating link prediction fairness beyond homophily. To support this, they formalize a useful taxonomy that aggregates both classical and flow-based structural bias measures.
2. The proposed extension to the BA model introduces explicit parameters for group balance, homophily, local community attachment, and degree variance, which facilitates systematic sweeps of graph topologies.
3. The study benefits from a synthetic corpus calibrated to real-world network parameters. Evaluating a wide range of both standard (GCN, Node2Vec) and fairness-aware baselines across this corpus improves the external validity of the results.
4. Using partial Spearman correlations combined with block-permutation tests is a methodologically sound approach to isolating structural effects independent of assortativity.

**Weaknesses**

1. The generator remains fundamentally tied to the BA model, meaning it lacks independent controls for realistic network properties such as clustering, triadic closure, or degree-corrected block heterogeneity. This limits the coverage of the synthetic graphs and introduces the risk of confounding variables among the bias dimensions.
2. The $R^2$ values approaching 1.0 in the fairness regressions strongly suggest potential data leakage or overfitting. The authors do not adequately explain their train/validation splits or provide robustness checks for these regressions.
3. The empirical evaluation relies strictly on Statistical Parity and Equal Opportunity. The authors omit rank-aware exposure metrics (e.g., NDKL) and distance-conditional (k-hop) fairness analyses, which are standard in modern recommendation evaluations.
4. There is a heavy reliance on synthetic data with limited real-world evaluation beyond initial calibration. Conclusions regarding generality would significantly benefit from applying the analysis directly to multiple real graphs with diverse topologies.

**Audience:**

Yes

**Audience Explanation:**

Fairness in graph neural networks and link prediction is a important area of research. Identifying that current fair methods frequently fail under varied topological constraints provides crucial insight for researchers designing the next generation of equitable and robust graph algorithms.

**Broader Impact Concerns:**

The authors provided a satisfactory Broader Impact Statement

**Claims And Evidence:**

Yes

**Claims Explanation:**

The authors systematically validate their synthetic generator against different networks to ensure the baseline topologies are realistic. The experimental protocol tests a massive grid of generated graphs, yielding results that are well-supported by statistical significance testing. This robustly rejects their null hypothesis (H3), providing clear evidence that biases beyond homophily inherently impact model fairness.

**Requested Changes:**

1. Explicitly detail the procedures used to prevent overfitting in the Random Forest regressions, such as cross-validation techniques or leave-one-configuration-out checks, and report out-of-distribution performance.
2. Incorporate exposure-aware fairness metrics (like NDKL) and k-hop fairness into the analysis to demonstrate whether the findings hold true for ranked link prediction tasks.
3. Report exact effect sizes for the partial correlations rather than relying solely on p-values, and apply multiple-testing corrections across the evaluated models and datasets.
4. Apply the analytical framework directly to multiple, diverse real-world graph datasets to substantiate the claims of generality derived from the synthetic corpus.